# Does the Elemental Composition of Rock Surfaces Affect Marine Benthic Communities of Diatoms and Cyanobacteria?

Anastasiia Blaginina [1], Daria Balycheva [1], Ekaterina Miroshnichenko [1], Larisa Ryabushko [1],
Sergey Kapranov [1], Sophia Barinova [2,*] and Denis Lishaev [1]

[1]  A.O. Kovalevsky Institute of Biology of the Southern Seas of RAS, 299011 Sevastopol, Russia;
    aablagini@ibss-ras.ru (A.B.); dashik8@gmail.com (D.B.); inerlim@gmail.com (E.M.);
    larisa.ryabushko@yandex.ru (L.R.); sergey.v.kapranov@yandex.ru (S.K.); chandler37@yandex.ru (D.L.)
[2]  Institute of Evolution, University of Haifa, Mount Carmel, Haifa 498838, Israel
[*]  Correspondence: sbarinova@univ.haifa.ac.il

**Abstract:** Rocky seabeds, as an integral part of ecotopes in marine ecosystems, are actively inhabited by diatoms and cyanobacteria. It is currently unknown whether the element contents in the surface layer of seabed rocks affect the epilithon species composition and abundance in microphytobenthos communities in the sea. The results of this study on the rock surface element composition and correlation analysis of the element contents with the abundance of epilithon diatoms and cyanobacteria in three bays in Sevastopol (Black Sea) are presented. Ca, Fe, and Si were the major elements with the largest weight fraction in the rock surface layer. Using cluster analysis, the differentiation of samples in the content and distribution of these three elements was shown. In total, 63 taxa of diatoms and 20 species of cyanobacteria were found, with their abundance ranging from 14,000 to 17,6000 cells/cm$^2$ and from 12,000 to 1,198,000 cells/cm$^2$, respectively. In general, it was found that the elemental composition of the rock surface is not a decisive factor affecting the total abundance of the benthic diatom and cyanobacterial communities as no strong correlations with any element contents were observed. However, when analyzing the abundance of populations of certain largely non-dominant species, the majority of diatoms showed noticeable ($r = 0.5–0.7$) to very high ($r = 0.9–0.99$) correlations with Fe. The highest positive correlations were noted for the diatoms *Bacillaria paxillifer* and *Navicula directa* with Fe. For the cyanobacteria *Chroococcus minutus*, *Pseudanabaena minima*, and *Spirulina subsalsa,* strong positive correlations with Ca and negative correlations with Si were observed. The correlations with Fe were very strong and negative for *Lyngbya confervoides* and strong and positive for *Kamptonema laetevirens* and *Phormidium holdenii*.

**Keywords:** diatoms; cyanobacteria; community; rocks; epilithon; elements; correlations

## 1. Introduction

Biogeochemical cycles in the sea, i.e., the cyclic fluxes of elements and inorganic compounds between the community and the ecotope, in which biogenic and chemogenic processes are coupled, are responsible for both physicochemical phenomena, such as mineralization and sedimentation, and for bioproduction processes [1]. Communities of cyanobacteria and diatoms dominate marine epilithon [2–6] and are actively involved in marine biogeochemical cycles, especially in biomineralization on substrates [7–13]. Some cyanobacterial species can biomineralize carbonates intracellularly [14].

The rocky seabed is an integral part of biotopes of marine ecosystems. According to the granulometric composition in shallow waters in the coastal zone, clastic sedimentary rocks of the psephitic structure predominate, which are represented mainly by monomineral conglomerates and loose rocks: crushed stone, pebbles, gravel, etc. In cemented rocks, the binders are carbonates (calcite, dolomite), silicon oxide (opal, chalcedony, quartz), iron oxides, clay minerals, and a number of others [15]. The shores of Sevastopol Bay are made

up of the Neogene, mainly Sarmatian, limestone and are categorized as resistant to abrasion. The lithological composition of the coast of the Sevastopol Bay is represented by layered, weakly recrystallized limestone with marl interlayers [16] predominantly of carbonate and quartz-carbonate composition [17]. The nature of rocky seabeds is one of the important abiotic factors in aquatic ecosystems for the life of attached hydrobionts, which, in turn, are involved in the formation of sedimentary rocks. From the geosphere, the biota receives the necessary minerals and returns the products of vital activity there. Organisms in unity with the natural geochemical environment are the object of study in geochemical ecology [18], and from this point of view, benthic communities are of particular interest. Species living on rocky substrates in the coastal area can be affected by many stress factors, such as intense solar radiation, drying and rehydration, extreme temperature fluctuations, lack of nutrients, etc. [19]. Geochemical inhomogeneity is one of the most important factors responsible for the variability in the metabolism and synthesis of biologically active compounds in living organisms. It affects the species diversity in communities, the structure of populations [18], and the benthic community formation, which are directly dependent on the structure and surface area of the inhabited substrates [4,13,20–22]. Biogeochemical cycles primarily involve nutrients such as Si, $NH_3$, $H_2S$, P, Fe, Mn, and Zn [23,24] that are necessary for the organisms' lives. Diatoms are the main consumers of silicate [25–27] and belong to "iron-loving" organisms [28–30]. Such an element as calcium promotes the secretion of polysaccharides, which ensure the attachment of cells to the substrate and the formation of colonies [31].

The chemical composition of marine rocks reflects the chemical composition of the sedimentation environment, i.e., sea water. On the other hand, vice versa, the seawater composition depends on the leaching of element ions from the rocky bed [25,29]. Thus, these factors are inextricably linked and make an important contribution to the state of the habitats of aquatic organisms. While the influence of hydrochemical conditions on microorganisms is a matter of continuous research, the effects of the chemical composition of the seabed are still obscure [26]. The impact of the substrate on benthic organisms is as significant as their participation in the processes of biomineralization and biodeposition. Biofilms and their constituents, extracellular polymeric substances, slowly change the substrate properties, including the pH at the surface (chemical vulnerability), grain cohesion (physical vulnerability), and the hydrophobicity [19]. Despite the abiotic stresses, cyanobacteria tend to populate even bare rock surfaces [32]. Diatoms and cyanobacteria play an important role in the biogeochemical cycles in marine coastal ecosystems. But the effects of substrate chemistry on the microphytobenthos communities on rocky beds in the seas and, in particular, in the Black Sea have not yet been studied. Given that the leaching processes are intensified under the action of attached communities [33], the chemical composition of the surface can be crucial in these processes.

The aim of this work is to study diatom and cyanobacterial communities on rocky seabeds in relation to the elemental composition of the rock surface in three bays in the Black Sea.

## 2. Materials and Methods

### 2.1. Sampling Area

The sampling of rocks for studying their elemental composition and the epilithon communities of diatoms and cyanobacteria on them was carried out in three bays: Martynova Bay (MB) (44°37′01.52″ N 33°30′21.37″ E), Yuzhnaya Bay (YuB) (44°36′24.57″ N 33°31′45.99″ E), and Inkerman Bay (IB) (44°36′50.22″ N 33°36′00.64″ E), shown in Figure 1. These are marginal areas of Sevastopol Bay, a typical semi-enclosed brackish water area of estuarine type in the Black Sea. It is located in the southwestern part of the Crimean peninsula (45°24′ N 33°00′ E), indented with a number of small bays containing well-developed port infrastructure, and involved in different economic activities in the city of Sevastopol. It is these water areas that are most prone to environmental disturbance due to anthropogenic pressure. In the eastern part (IB), the bay receives the waters of the Chernaya River, which

belong to the hydrocarbonate class, the calcium group. From the west (MB), sea waters enter the bay. The main factor that drives the water circulation in the bay is wind. Under its action, the change in the spatial distribution of temperature and salinity in the bay occurs within a few hours. Thus, the intensity of water exchange is determined mainly by surge phenomena [34]. The average silicate fluctuations in Sevastopol Bay are between 0.6 and 13.1 μM and the highest concentrations of silicon and other nutrients are typical for YuB. The pH value of the bay water is 8.3–8.4. In the IB area, the water contains high concentrations of iron.

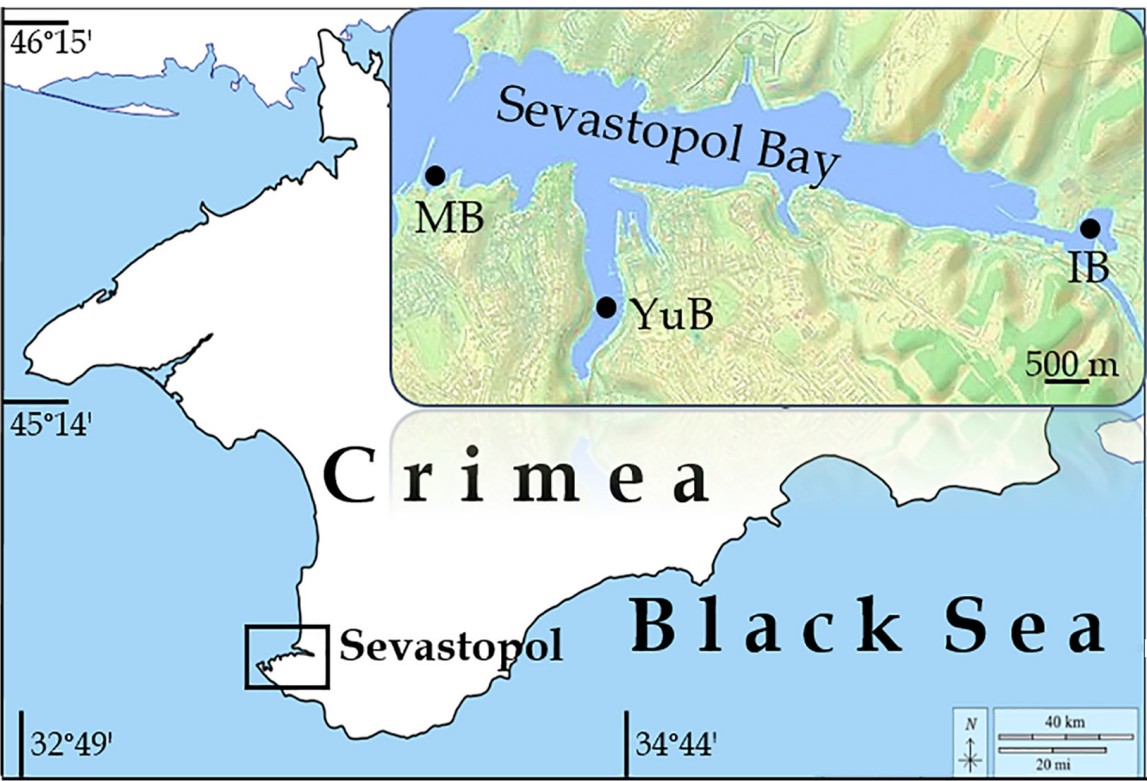

**Figure 1.** Map of the sampling area (Sevastopol Bay, Black Sea): MB = Martynova Bay, YuB = Yuzhnaya Bay, IB = Inkerman Bay.

Seabed rock samples were collected on 2 September 2019 simultaneously from the three bays at the average water temperature in Sevastopol Bay of 22 °C and salinity ranging from 16 (IB) to 18 psu (MB). In MB and IB, rock samples were taken randomly from a depth of 1–2 m, and in YuB, the sampling was carried out at a depth of 6 m by diving. From each sampling station, one liter of seawater was taken for the sample processing.

*2.2. Geological Samples*

The characteristics of sedimentary rocks in the sampling areas are as follows. The rocky bed of Inkerman Bay is represented by nummulite and bryozoan limestones. Oolitic spongy limestone is typical for Yuzhnaya Bay, and nummulite limestone is typical for Martynova Bay [35].

The rock samples differed in surface area, ranging from 18 to 620 cm$^2$, and in surface structure, as shown in Figure 2a–f for the samples from Martynova Bay. The surface area of rocks was calculated using the equation [36] $S = \pi/3 \, (xy + yz + xz)$, where x, y, and z are the linear sizes of the substrate. The surface areas of each rock sample were as follows: $S_{IB}$ = 25, 19, 29, 18, and 23 cm$^2$; $S_{MB}$ = 102, 69, and 77 cm$^2$; and $S_{YuB}$ = 620, 330, and 280 cm$^2$, with the determination error being $\pm 5\%$.

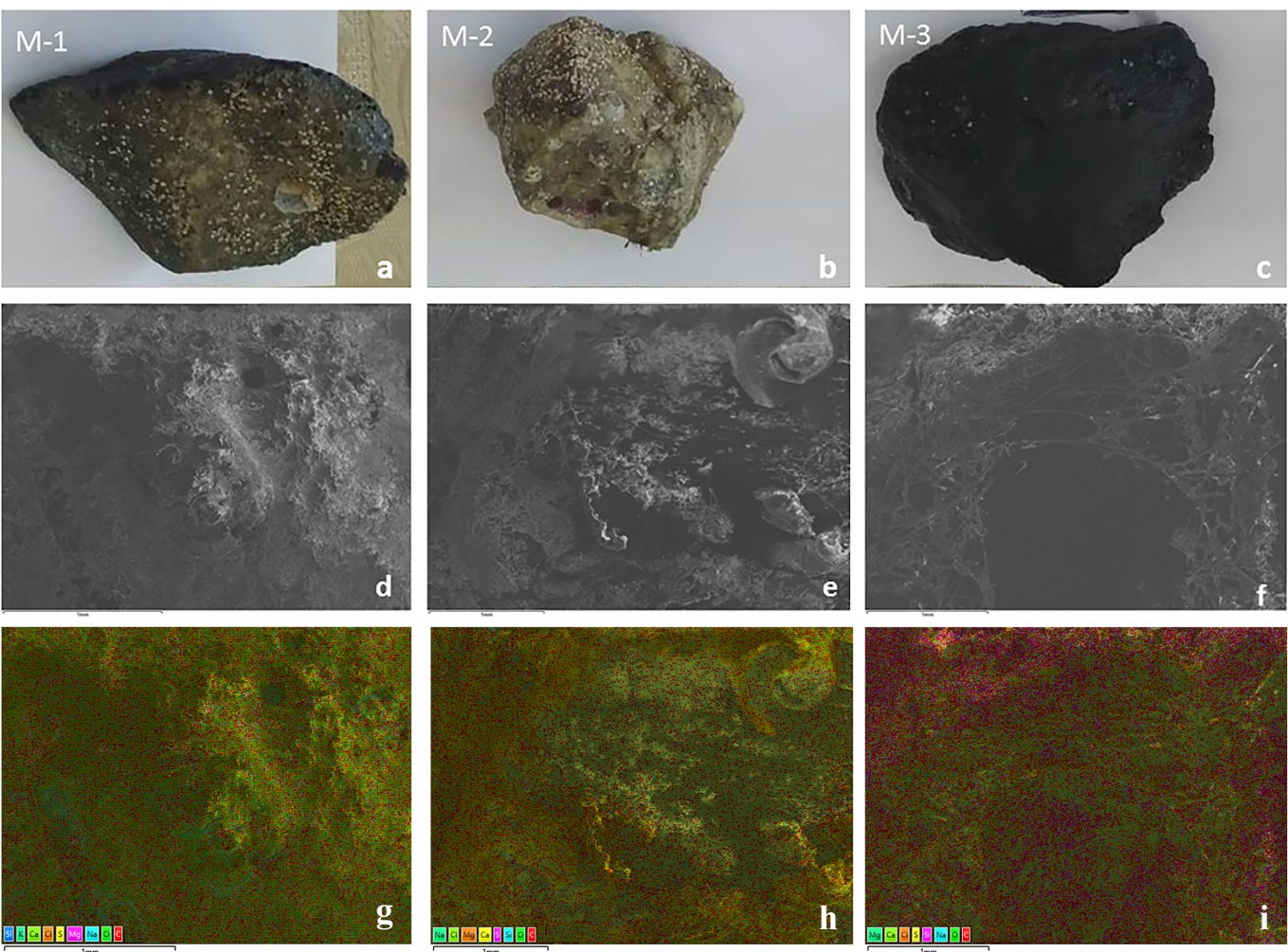

**Figure 2.** Rocky substrates from Martynova Bay (MB-1; MB-2; MB-3): (**a–c**) general appearance; (**d–f**) SEM image of the rock surface; (**g–i**) backscattered electron images (SEM) of the elemental composition of rock surface.

To characterize the elemental composition and determine the origin of the rocky beds, an energy-dispersive X-ray spectroscopy (EDX) system incorporated into a scanning electron microscope (SEM) Hitachi SU3500 (Japan) was used. Gold-palladium coatings were sputtered on rock samples in a Leica EM ACE200. The element contents in the rock surface layer were measured in 10 spots on each rock sample.

The origin of the rocky substrates was identified according to the method for determining the genesis of marine sedimentary deposits [15]. To this end, the rocks were placed in containers and stored in a dry dark room for 3–6 months. In the process of the evaporation of solutions coming to the surface of rocks from the pores, seasonal minerals are formed [37]. Then, the element composition of seasonal minerals formed on the rock surface was analyzed, and it was the criterion for deciding on the marine or terrestrial genesis of the rock surface. The element abundance order Na > Mg > Ca in the seasonal minerals, similar to that in seawater, suggests the marine origin of the rocks under investigation [15].

### 2.3. Microphytobenthos Sampling

Diatoms and cyanobacteria were sampled from the rock surface by scrubbing with a plastic brush and washing out with filtered seawater into containers for analysis. The collected samples were fixed with 4% formalin for further treatment. All samples were examined under the light microscopes (LMs) Bresser Bio Science Trino and Axioskop 40 C. Zeiss at the appropriate magnifications of 10 × 20, 10 × 40, and 10 × 100. For the more

accurate identification of diatoms, scanning electron microscopy (SEM, Hitachi SU3500, Japan) was applied. The sample preparation for the examination was carried out according to [38]. The diatom suspension was cleaned of organic matter by keeping it in $KMnO_4$ for 24 h, which was followed by adding HCl and heating this mixture to remove insoluble salts (e.g., carbonates). Then, the samples were rinsed with distilled water using repeated centrifugations to remove acid. Dried preparations of diatom valves were coated with gold-palladium for the SEM visualization.

The identification of diatoms and cyanobacteria was carried out using identification guides and atlases [39–49]. The occurrence of cyanobacterial species in a sample was assessed visually and evaluated using a 5-point scale from 1 (very rare) to 5 (very frequent) [50].

Cells of diatoms and cyanobacteria were counted under the microscope in a hemocytometer (Goryaev's chamber) in three replicates for each sample. The abundance ($N$, cell/cm$^2$) was calculated using the following relationship [4]:

$$N = \frac{n \cdot V}{S \cdot Vk} \tag{1}$$

where $n$ is the number of cells in Goryaev's chamber; $V$ is the sample volume in mL; $Vk = 0.9$ mm$^3$ is the volume of Goryaev's chamber; and $S$ is the surface area of the rock.

*2.4. Statistical Analysis*

Statistical data processing was performed using Microsoft Office Excel software. The GRAPHS software module was used to perform cluster analysis with the Sörensen coefficient [51] as the species similarity measure. Correlation analysis was carried out in the R programming language in the open-access R-studio development environment.

The degree of association between the average abundances of diatoms and cyanobacteria was based on the linear correlation coefficient ($r$). The linear correlation coefficient was also calculated for the total abundance of diatom and cyanobacterial communities and the elemental composition. To assess the relationship between the abundance of individual species and the element contents, Spearman's correlation coefficient ($r_s$) was used. Only values falling within the 95% confidence interval were taken into account. The degree of association was defined according to the following correlation coefficient grades: 0.1–0.3 (weak), 0.3–0.5 (moderate), 0.5–0.7 (noticeable), 0.7–0.9 (strong), and 0.9–0.99 (very strong).

**3. Results**

*3.1. Abiotic Data Analysis*

3.1.1. Elemental Composition of Rock Surfaces

The elemental composition and distribution in the rock surface layer was studied. The results of the EDX study, shown in Figures 2 and 3 and in Table 1, allowed us to establish that elements are unevenly distributed in the rock surface layer. The major elements with the highest weight fraction were Ca, Fe, and Si (Figure 3). The example of MB shows that the content (in %) of these three elements on the rock surfaces in this bay varied in wide ranges: Si from 0.4 to 6.3, Ca from 0.3 to 19.5, and Fe from 0 to 0.9 (Table 1, Figures 3 and 4). In IB, the Fe content was 3–18 times higher than in YuB, and in MB rocks, this element was virtually absent. The weight fraction of O and C was high, too (Figure 3). These elements are constituents of compounds, including such biogenic components of sedimentary rocks as carbonates ($CaCO_3$) and oxides ($SiO_2$).

Differences in the Fe, Ca, and Si content on the rocks' surfaces were present not only in different bays but even within the same station and may be due to the physical and geographical characteristics of Sevastopol Bay, which is characterized by an indented coastline, as shown in Figure 1. Based on the elemental composition of the sampled rocks, the following conclusion was made about their origin: out of eleven samples under study, ten were of continental origin, and only one MB-3* rock sample was of marine origin

(Table 1) since the inequality Na > Mg > Ca was observed in the elemental composition of its surface, namely 1.9 > 0.7 > 0.3 (Figure 4 and Table 1).

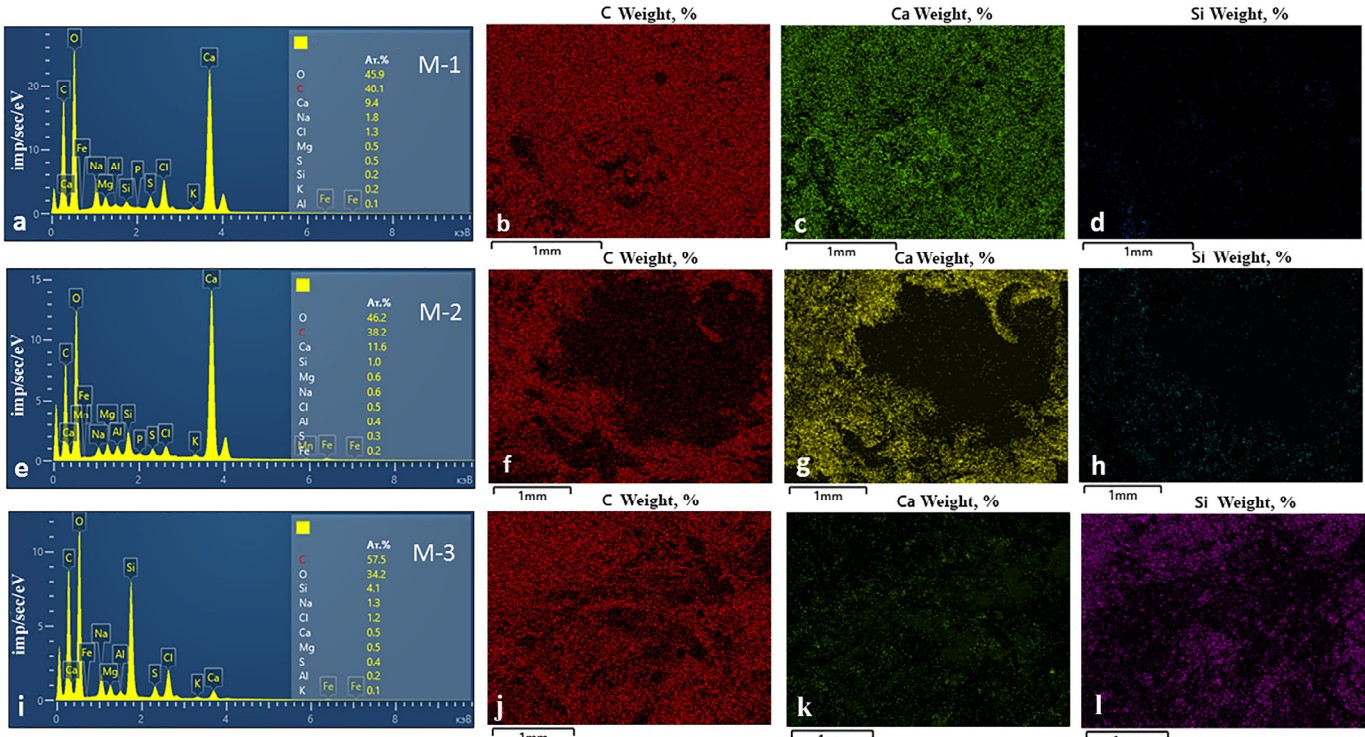

**Figure 3.** Analysis of the elemental composition (%) of three rock surfaces from Martynova Bay (MB-1, MB-2, MB-3) using EDX: (**a**,**e**,**i**) EDX spectra; (**b**–**d**, **f**–**h**, **j**–**l**) backscattered electron images (SEM) of distribution of the major elements in the rock surface layer ((**b**,**f**,**j**) = C, (**c**,**g**,**k**) = Ca, (**d**,**h**,**l**) = Si).

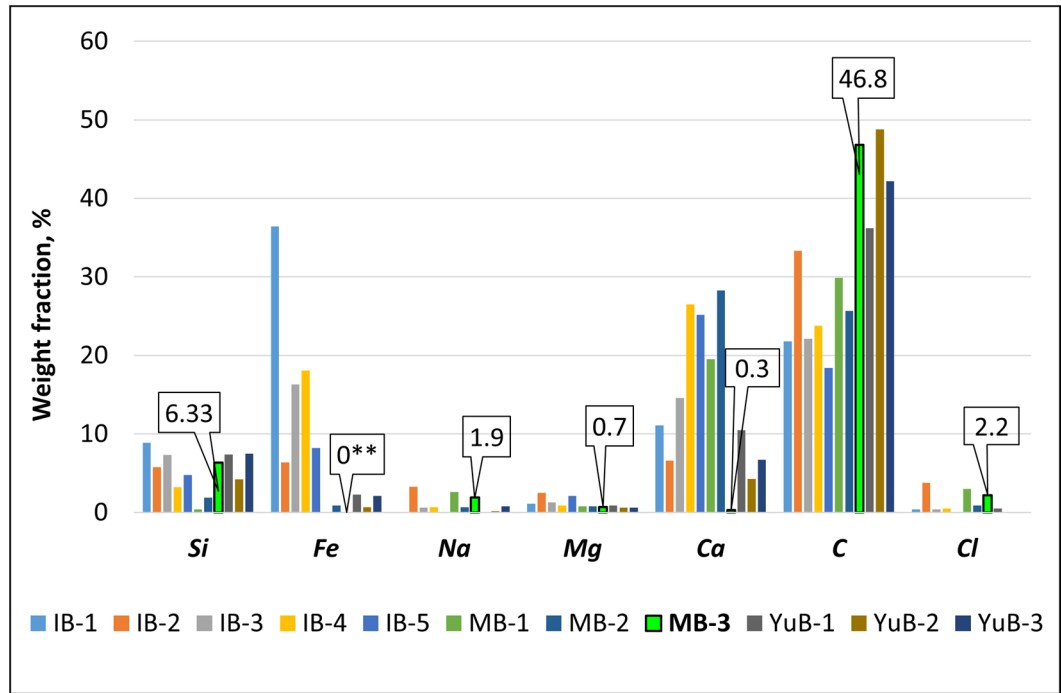

**Figure 4.** Element distribution in the surface layer of rocks from three bays in Sevastopol. IB: Inkerman Bay, YuB: Yuzhnaya Bay, MB: Martynova Bay. The MB-3* rock sample is of marine origin. ** Below limit of detection.

**Table 1.** Contents of major elements, W, % of rock surfaces of continental (C) and marine (M) origin in three bays in Sevastopol.

| Rock Sample | Si | Fe | Na | Mg | Ca | C | Cl | Origin |
|---|---|---|---|---|---|---|---|---|
| | | | Inkerman Bay (IB) | | | | | |
| IB-1 | 8.9 ± 0.05 | 36.4 ± 0.1 | – | 1.1 ± 0.01 | 11.1 ± 0.05 | 21.8 ± 0.1 | 0.4 ± 0.001 | C |
| IB-2 | 5.8 ± 0.05 | 6.4 ± 0.05 | 3.3 ± 0.01 | 2.5 ± 0.01 | 6.6 ± 0.01 | 33.3 ± 0.1 | 3.8 ± 0.01 | C |
| IB-3 | 7.3 ± 0.01 | 16.3 ± 0.1 | 0.6 ± 0.001 | 1.3 ± 0.01 | 14.6 ± 0.1 | 22.1 ± 1.2 | 0.4 ± 0.001 | C |
| IB-4 | 3.2 ± 0.1 | 18.1 ± 0.05 | 0.7 ± 0.01 | 0.9 ± 0.05 | 26.5 ± 0.1 | 23.8 ± 0.1 | 0.5 ± 0.001 | C |
| IB-5 | 4.8 ± 0.1 | 8.2 ± 0.1 | – | 2.1 ± 0.01 | 25.2 ± 0.05 | 18.4 ± 0.05 | – | C |
| | | | Martynova Bay (MB) | | | | | |
| MB-1 | 0.4 ± 0.001 | – | 2.6 ± 0.01 | 0.8 ± 0.001 | 19.5 ± 0.1 | 29.9 ± 0.05 | 3.0 ± 0.03 | C |
| MB-2 | 1.87 ± 0.1 | 0.9 ± 0.01 | 0.7 ± 0.01 | 0.8 ± 0.001 | 28.3 ± 0.1 | 25.7 ± 0.1 | 0.9 ± 0.01 | C |
| MB-3 | 6.33 ± 0.1 | – | 1.9 ± 0.01 | 0.7 ± 0.03 | 0.3 ± 0.03 | 46.8 ± 0.1 | 2.2 ± 0.1 | M |
| | | | Yuzhnaya Bay (YuB) | | | | | |
| YuB-1 | 7.4 ± 0.1 | 2.3 ± 0.1 | – | 0.9 ± 0.1 | 10.5 ± 0.1 | 36.2 ± 0.2 | 0.5 | C |
| YuB-2 | 4.2 ± 0.1 | 0.7 ± 0.1 | 0.2 ± 0.1 | 0.6 ± 0.1 | 4.3 ± 0.1 | 48.8 ± 1.2 | – | C |
| YuB-3 | 7.5 ± 0.1 | 2.1 ± 0.1 | 0.8 ± 0.1 | 0.6 ± 0.1 | 6.7 ± 0.1 | 42.2 ± 0.01 | – | C |

"–" element contents below limit of detection.

### 3.1.2. Cluster Analysis of the Rocks Similarity in Terms of Composition and Content of Chemical Elements on Their Surface

Cluster analysis (Figure 5a) of the element contents in the rock surface layer demonstrated a similarity of more than 68% among the clusters irrespective of the sampling area. Rocks from the same sampling stations were combined in clusters with a similarity of more than 80%, which reflects the influence of similar environmental conditions.

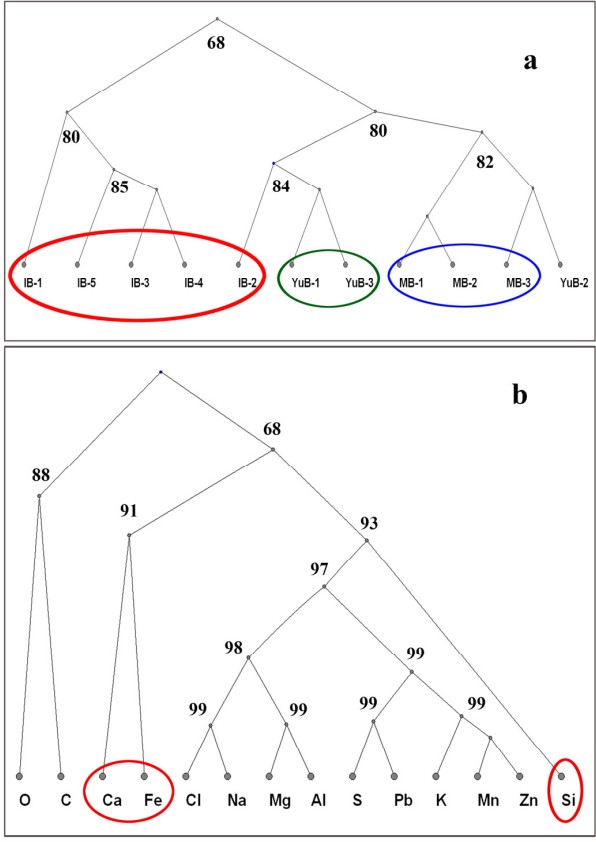

**Figure 5.** Dendrograms of the similarity (in %) of the element contents in rocky beds based on the Sörensen coefficient: (**a**) elemental composition similarity, red—Inkerman Bay, green—Martynova Bay, blue—Yuzhnaya Bay; (**b**) similarity of the distribution of elements in the rock surface layer.

A dendrogram of similarity in the element contents and distribution in rocks is presented in Figure 5b. Ca and Fe are isolated in a separate cluster with a high similarity. Another separate cluster that combined C and O indicates the presence of either a biotic component or carbonates on the rock surface [48–50]. It should be noted that Si formed its own singleton cluster, which is 93% similar to that of other elements and has a similarity of 68% with the cluster of Ca and Fe. The high similarity and contents of these lithogenic elements determined the need for evaluating their impact on the diatom and cyanobacterial communities in the rock epilithon. The data obtained show the geochemical inhomogeneity of the habitat of benthic organisms, which inevitably affects the flows of matter, metabolism, and adaptation of organisms to specific geochemical conditions and is reflected in the mosaic pattern of epilithic communities.

### 3.2. Biotic Data Analysis

3.2.1. Composition of the Diatom and Cyanobacterial Communities on the Rock Surface

The species composition and abundance of diatoms and cyanobacteria were estimated in the epilithon of rocks from the three bays. In Table 2, a total of 63 identified diatom taxa are presented, with 31 of them being in Martynova Bay, 48 in Inkerman Bay, and 19 in Yuzhnaya Bay. Among them, eight species were common. Marine (26) and brackish-marine (18) species were dominant among diatoms. There were a few freshwater (seven) and freshwater-brackish (three) species, and twenty-eight cosmopolite species were detected. In MB, the abundance varied from 50,000 to 176,000 cells/cm$^2$; in IB, it varied from 51,000 to 149,000 cells/cm$^2$; and in YuB, it ranged from 14,000 to 28,000 cells/cm$^2$ (Table 3).

Among cyanobacteria, 20 species were found which belong to 6 orders, 10 families, and 12 genera. Of these, fifteen species were found in Martynova Bay, seven in Inkerman Bay, and four in Yuzhnaya Bay. The highest species richness saturation was noted for the family Oscillatoriaceae (nine). Most of the species were marine (six) and halotolerant (eight), with the cosmopolite (ten) and boreal–tropical–notal (six) occurrence (Table 2). Some species of diatoms and cyanobacteria examined under the optical and scanning electron microscopes are presented in Figures 6 and 7.

Eight diatom taxa, *Berkeleya rutilans* (Figure 6A,B), *Undatella lineolata* (Figure 6C), *Halamphora coffeiformis* (Figure 6D,E), *Cylindrotheca closterium* (Figure 6G), *Psammodictyon panduriforme* var. *minor* (Figure 6I,J), *Navicula ramosissima* (Figure 6M), *Nitzschia sigma* (Figure 6N), and *Haslea subagnita* (Figure 6O), and one cyanobacterium, *Leptolyngbya fragilis* (Figure 7C), were common to all bays. The numbers of species of diatoms/cyanobacteria that occurred in only one of the bays were 10/0 in YuB, 23/5 in IB, and 8/10 in MB.

The total abundance of diatoms varied from 14,000 to 176,000, and the total abundance of cyanobacteria varied from 12,000 to 1,198,000 cells/cm$^2$ (Tables 2 and 3). On the rock of marine origin (MB-3*), the highest abundance and number of marine diatom species (12) were noted. However, as shown in Table 3, the abundance of cyanobacteria on MB-3* was low (249,000 cells/cm$^2$), and the number of cyanobacterial species was the smallest. The maximum abundance of cyanobacteria was noted on rocks of continental origin in Inkerman Bay (IB-3). The minimum abundance of diatoms and cyanobacteria was found on all rocks from Yuzhnaya Bay.

In the epilithon of MB, 31 diatom taxa were found, which belong to 23 genera (Table 2). The average abundance was 97,000 cells/cm$^2$, with the dominance of the bentoplanktonic diatom *Cylindrotheca closterium* (Figure 6F,G) with $N$ = 18,000 cells/cm$^2$. The subdominant species were the small-celled *Berkeleya rutilans* (Figure 6A,B) and *Parlibellus delognei* (Figure 6H) with 12,000 and 14,000 cells/cm$^2$, respectively. Cyanobacteria were represented by 15 species, with the dominance of *Lyngbya confervoides* and *Potamolinea aerugineocaeruleum* (Table 2). The total abundance of cyanobacteria varied from 74,760 to 380,740 cells/cm$^2$.

**Table 2.** Species composition and abundance (*N*, $10^3$ cells/cm$^2$) of Bacillariophyta and occurrence (F) of Cyanobacteria in epilithon of the three bays and their ecological (Ecol) and phytogeographical (Phyt) characteristics with abbreviations.

| Taxa | MB | IB | YuB | Ecol | Phyt | Abbreviated Name |
|---|---|---|---|---|---|---|
| **Phylum Bacillariophyta** | | | | | | |
| *Achnanthes brevipes* C.A. Agardh 1824 var. *brevipes* | 0.598 | – | – | BM | C | AchBr |
| *Achnanthes brevipes* var. *intermedia* (Kützing) P.T. Cleve 1895 | 0.924 | 3.654 | – | BM | C | AchBrI |
| *Amphora bigibba* Grunow ex A. Schmidt 1875 | 2.643 | – | 2.311 | M/F | BT not | AmBig |
| *Amphora helenensis* Giffen 1973 | – | + | – | M | B not | |
| *Amphora ovalis* (Kützing) Kützing 1844 | 0.990 | – | – | FB | C | AmOv |
| *Amphora pediculus* (Kützing) Grunow 1875 | – | + | – | F | ABT not | |
| *Aulacoseira granulata* (Ehrenberg) Simonsen 1979 | 2.815 | – | – | FB | C | AuGr |
| *Bacillaria paxillifer* (O.F. Müller) N. Hendey 1954 | – | 5.845 | 0.916 | BM | C | |
| *Berkeleya rutilans* (Trentepohl ex Roth) Grunow 1880 | 12.352 | 14.875 | 2.093 | BM | C | BerRu |
| *Caloneis liber* (W. Smith) P.T. Cleve 1894 | 0.881 | 0.335 | – | M | C | CaLib |
| *Carinasigma rectum* (Donkin) G. Reid 2012 | + | – | – | M | BT not | |
| *Cocconeis peltoides* Hustedt 1939 | – | + | – | M | B not | |
| *Cocconeis placentula* var. *euglypta* (Ehrenberg) P.T. Cleve 1895 | – | + | – | M | C | |
| *Cocconeis scutellum* Ehrenberg 1838 | + | 6.594 | – | BM | C | CoScu |
| *Cocconeis neodiminuta* Krammer 1990 | – | + | – | F | BT not | |
| *Cocconeis neothumensis* Krammer 1990 | – | + | – | M/F | B not | |
| *Coscinodiscus jonesianus* (Greville) Ostenfeld 1915 | 5.793 | 0.491 | – | M | B | CosJo |
| *Cylindrotheca closterium* (Ehrenberg) Reimer et Lewin 1964 | 18.380 | 1.021 | 4.449 | M | C | CyClo |
| *Cymbella helvetica* Kützing 1844 | + | – | – | F | BT not | |
| *Diploneis smithii* (Brébisson) P.T. Cleve 1894 | – | 2.746 | 0.916 | BM | C | |
| *Fallacia tenera* (Hustedt) D.G. Mann 1990 | – | + | – | F | B not | |
| *Gomphonemopsis pseudexigua* (Kützing) L.K. Medlin 1986 | 1.262 | + | – | M | ABT not | |
| *Grammatophora marina* (Lyngbye) Kützing 1844 | 1.728 | – | – | M | C | GrMa |

**Table 2.** *Cont.*

| Taxa | MB | IB | YuB | Ecol | Phyt | Abbreviated Name |
|---|---|---|---|---|---|---|
| *Gyrosigma tenuissimum* (W. Smith) J.W. Griffith et Henfrey 1856 | – | + | – | M | BT not | |
| *Halamphora coffeiformis* (C. Agardh) Levkov 2009 | 3.263 | 7.116 | 0.131 | BM | C | HaCof |
| *Halamphora hyalina* (Kützing) Rimet et R. Jahn 2018 | + | – | + | BM | ABT not | |
| *Haslea ostrearia* (Gaillon) Simonsen 1974 | – | 7.616 | 0.654 | M | BT | HOst |
| *Haslea subagnita* (Proschkina-Lavrenko) I.V. Makarova et N.I. Karajeva 1985 | 4.275 | + | 0.218 | B | C | HSub |
| *Licmophora abbreviata* C. Agardh 1831 | 1.332 | – | – | M | C | LiAb |
| *Licmophora communis* (Heiberg) Grunow 1880 | – | 2.910 | – | M | AB not | LicCom |
| *Licmophora paradoxa* (Lyngbye) C. Agardh 1828 | – | + | – | M | C | |
| *Nanofrustulum shiloi* (J.J. Lee, Reimer et McEnery) Round, Hallsteinsen et Paasche 1999 | – | + | – | M | B not | |
| *Navicula cancellata* Donkin 1873 | – | 3.047 | – | B | C | |
| *Navicula directa* (W. Smith) Ralfs ex Pritch. 1861 | – | 2.862 | 0.436 | BM | C | NavDir |
| *Navicula palpebralis* Brébisson ex W. Smith 1853 | 0.990 + 0.374 | 2.991 | – | M | ABT not | NavPal |
| *Navicula pennata* A.W.F. Schmidt 1876 | 3.167 | + | – | BM | BT not | NavPen |
| *Navicula perminuta* Grunow 1880 | – | + | – | M/F | C | |
| *Navicula ramosissima* (C. Agardh) P.T. Cleve 1895 | 3.291 | 3.815 | 1.047 | BM | ABT not | NavRam |
| *Navicula salinicola* Hustedt 1939 | – | + | – | F/BM | BT not | |
| *Nitzschia aequorea* Hustedt 1939 | – | + | – | M | BT not | |
| *Nitzschia dissipata* (Kützing) Rabenhorst 1860 | – | + | – | F | ABT not | |
| *Nitzschia frustulum* (Kützing) Grunow 1880 | – | + | – | M/F | C | |
| *Nitzschia hybrida* f. *hyalina* Proschkina-Lavrenko1963 | + | – | – | BM | B | |
| *Nitzschia inconspicua* Grunow 1862 | – | + | – | F | C | |
| *Nitzschia sigma* (Kützing) W. Smith 1853 | 0.761 | 0.363 | 0.523 | B | C | NitSigm |
| *Nitzschia sigmoidea* (Nitzsch) W. Smith 1853 | – | – | 0.218 | FB | BT not | NitSigd |
| *Parlibellus delognei* (Van Heurck) E.J. Cox 1988 | 13.889 | + | – | M | C | ParDel |

**Table 2.** *Cont.*

| Taxa | MB | IB | YuB | Ecol | Phyt | Abbreviated Name |
|---|---|---|---|---|---|---|
| *Plagiogramma staurophorum* (W. Gregory) Heiberg 1863 | – | 0.452 | – | M | BT not | PlSta |
| *Plagiotropis lepidoptera* (W. Gregory) Kuntze 1898 | + | + | – | M | ABT not | |
| *Pleurosigma elongatum* W. Smith 1852 | – | + | – | BM | C | |
| *Psammodictyon panduriforme* var. *minor* (Grunow) L.I. Ryabushko 2006 | 0.163 | + | 1.221 | M | BT not | PsPM |
| *Rhopalodia gibba* (Ehrenberg) O.F. Müller 1895 | 1.782 | – | – | F | ABT not | |
| *Rhopalodia gibberula* (Ehrenberg) O.F. Müller 1895 | 0.259 | – | 1.919 | M/F | BT not | RapGib |
| *Rhoicosphenia marina* (Kützing) M. Schmidt 1899 | – | 0.335 | – | M | B not | RhMar |
| *Seminavis ventricosa* (Gregory) M. Garcia-Baptista 1993 | 0.114 | 0.920 | – | M | C | SemVen |
| *Striatella unipunctata* (Lyngbya) C.A. Agardh 1832 | – | 2.174 | – | M | BT not | StUni |
| *Tabularia fasciculata* (C.A. Agardh) D.M. Williams et Round 1986 | – | 0.648 | 0.393 | BM | C | TabFa |
| *Tabularia parva* (Kützing) D.M. Williams et Round 1986 | – | + | – | BM | ABT not | |
| *Tabularia tabulata* (C.A. Agardh) Snoeijs 1990 | – | – | 0.480 | BM | C | TabTa |
| *Tetramphora decussata* (Grunow) Stepanek et Kociolek 2016 | – | + | – | M | BT not | |
| *Trachyneis aspera* (Ehrenberg) P.T. Cleve 1894 | 9.967 | 14.341 | – | M | C | TrAsp |
| *Tryblionella coarctata* (Grunow) D.G. Mann 1990 | – | 5.675 | 0.436 | BM | BT | TryCoar |
| *Undatella lineolata* (Ehrenberg) L.I. Ryabushko 2006 | 5.189 | 21.615 | 0.305 | BM | ABT not | UnLi |
| Average number of species | 31 | 48 | 19 | | | |
| Average abundance of species, $10^3$ cells/cm$^2$ | 97 | 112 | 19 | | | |
| **Phylum Cyanobacteria** | | | | | | |
| *Calothrix parva* Ercegovic 1925 | – | 1 | – | M | AB | Cp |
| *Chroococcus minutus* (Kützing) Nägeli 1849 | 2 | – | – | F | C | Cm |
| *Gloeocapsopsis crepidinum* (Thuret) Geitler ex Komárek 1993 | – | 1 | – | M/F | C | Gc |
| *Kamptonema laetevirens* (H.M. Crouan et P.L. Crouan ex Gomont) Strunecký, Komárek et J. Smarda 2014 | 3 | 4 | – | M/F | BT not | Kl |
| *Komvophoron breve* (N. Carter) Anagnostidis 2001 | 3 | – | – | B | BT | Kb |
| *Leptolyngbya fragilis* (Gomont) Anagnostidis et Komárek 1988 | 2 | 1 | 1 | M | C | Lpf |

**Table 2.** *Cont.*

| Taxa | MB | IB | YuB | Ecol | Phyt | Abbreviated Name |
|---|---|---|---|---|---|---|
| *Leptolyngbya subtilis* (West) Anagnostidis 2001 | 2 | – | 2 | F | BT not | Lps |
| *Lyngbya confervoides* C. Agardh ex Gomont 1892 | 4 | – | 1 | M | C | Lc |
| *Lyngbya martensiana* Meneghini ex Gomont 1892 | 3 | – | – | EuH | C | Lm |
| *Lyngbya semiplena* J. Agardh ex Gomont 1892 | 1 | – | – | M/F | BT not | Ls |
| *Oscillatoria bonnemaisonii* P. Crouan et H. Crouan ex Gomont 1892 | 1 | – | – | M | BT not | Ob |
| *Oscillatoria corallinae* Gomont ex Gomont 1890 | – | 1 | – | M | BT not | Oc |
| *Phormidium breve* (Kützing ex Gomont) Anagnostidis et Komárek 1988 | 1 | – | – | F | C | Phb |
| *Phormidium holdenii* (Forti) Branco, Sant'Anna, Azevedo et Sormus 1997 | – | 4 | – | M/F | ABT not | Phh |
| *Phormidium nigroviride* (Thwaites ex Gomont) Anagnostidis et Komárek 1988 | 3 | – | – | M | C | Phn |
| *Phormidium retzii* Kützing ex Gomont 1892 | – | 2 | – | F | C | Phr |
| *Potamolinea aerugineacaeruleum* (Gomont) M.D. Martins et L.H.Z. Branco 2016 | 4 | – | 1 | F | C | Pa |
| *Pseudanabaena minima* (G.S. An) Anagnostidis 2001 | 2 | – | – | F | ABT not | Psm |
| *Spirulina subsalsa* Oersted ex Gomont 1892 | 1 | – | – | M/F | C | Ss |
| *Spirulina subtilissima* Kützing ex Gomont 1892 | 2 | – | – | M/F | BT not | Ssm |
| Total number of species: 20, of which in bays: | 15 | 7 | 4 | | | |

Note: (–) the species was absent; (+) the species was present in sample, but was not counted. Relation to water salinity (Ecol): M = marine, B = brackish, BM = brackish-marine, FW = freshwater, FB = freshwater-brackish, EuH = euryhaline species. Phytogeographical characteristics (Phyt): B = boreal, AB = arctic–boreal, BT = boreal–tropical, ABT = arctic–boreal–tropical, C = cosmopolite, not = notal species found in the southern hemisphere.

**Table 3.** Rock origin and abundances (*N*, $10^3$ cells/cm$^2$) of diatoms and cyanobacteria in the rock epilithon from three bays.

| Rock Sample | Rock Origin | Diatoms | Cyanobacteria |
|:---:|:---:|:---:|:---:|
| IB-1 | C | 104 | 588 |
| IB-2 | C | 149 | 899 |
| IB-3 | C | 140 | 1198 |
| IB-4 | C | 118 | 297 |
| IB-5 | C | 51 | 158 |
| MB-1 | C | 50 | 381 |
| MB-2 | C | 65 | 75 |
| MB-3 | M | 176 | 249 |
| YuB-1 | C | 28 | 42 |
| YuB-2 | C | 14 | 35 |
| YuB-3 | C | 18 | 12 |

In IB, diatoms were represented by 48 taxa from 30 genera. Their average abundance was 112,000 cells/cm$^2$, with the dominance of *Undatella lineolata* (Figure 6C) at 22,000 cells/cm$^2$. The subdominants were *Berkeleya rutilans* and *Trachyneis aspera*, with abundances of 15,000 and 14,000 cells/cm$^2$, respectively. Here, seven cyanobacterial species were found, with the dominance of *Kamptonema laetevirens* and *Phormidium holdenii* (Figure 7B). The average total abundance was 628,000 cells/cm$^2$, ranging from 158,000 to 119,8000 cells/cm$^2$ (Table 3).

In YuB, 19 taxa of diatoms belonging to 14 genera were found. The diatom abundance was 19,000 cells/cm$^2$, with the dominance of *Cylindrotheca closterium* (4000 cells/cm$^2$), *Amphora bigibba* (Figure 6K,L), and *Berkeleya rutilans* (*N* = 2000 cells/cm$^2$). There were only four species of cyanobacteria in YuB, with the average total abundance of 29,780 cells/cm$^2$, which was the smallest in comparison with the other bays.

The quantitative characteristics of the benthic communities of diatoms and cyanobacteria were used in a cluster analysis of the structure of communities on different rocks, shown in Figure 8. Communities on rocks from the same station were allocated in clusters with a similarity above 80%, and the similarity of the species composition and abundance between the areas was 31–39%.

Consequently, the dendrogram of the rock similarity in terms of the elemental composition (Figure 5a) is almost identical to that of the community composition on the same rocks (Figure 8). However, the similarity of rocks between water areas was high, from 68 to 80%, and the similarity of benthic communities between water areas was low, only 31–39%.

3.2.2. Correlations between the Abundance of Diatoms and Cyanobacteria in Epilithon Communities

Since diatoms and cyanobacteria form the basis of benthic communities, the biocenotic relationships between them are of particular interest. The correlation analysis showed a positive significant relationship between their abundances, shown in Figure 9. Similar responses of the photosynthetic microorganisms to environmental factors indicate their joint participation in the formation of benthic communities.

*3.3. Biotic Factors in Relation to the Abiotic Variables*

In the correlation analysis of the total average abundance of the diatom and cyanobacterial communities and the element contents in the rock surface layer for each rock, the correlation coefficient, shown in the Table 4, varied mostly in the ranges characteristic of weak and moderate associations (from −0.01 to ±0.5), which indicate the absence of a reliable correlation. When assessing the correlations of the population abundances with all detected elements, they showed considerable variability among sampling areas. The elemental composition of the rocks' surfaces was similar across regions (Figure 5a), and therefore, the abundance of diatoms and cyanobacteria was more strongly influenced by other factors, e.g., substrate surface structure, water temperature, depth, illuminance, and grazing by phytophages [4].

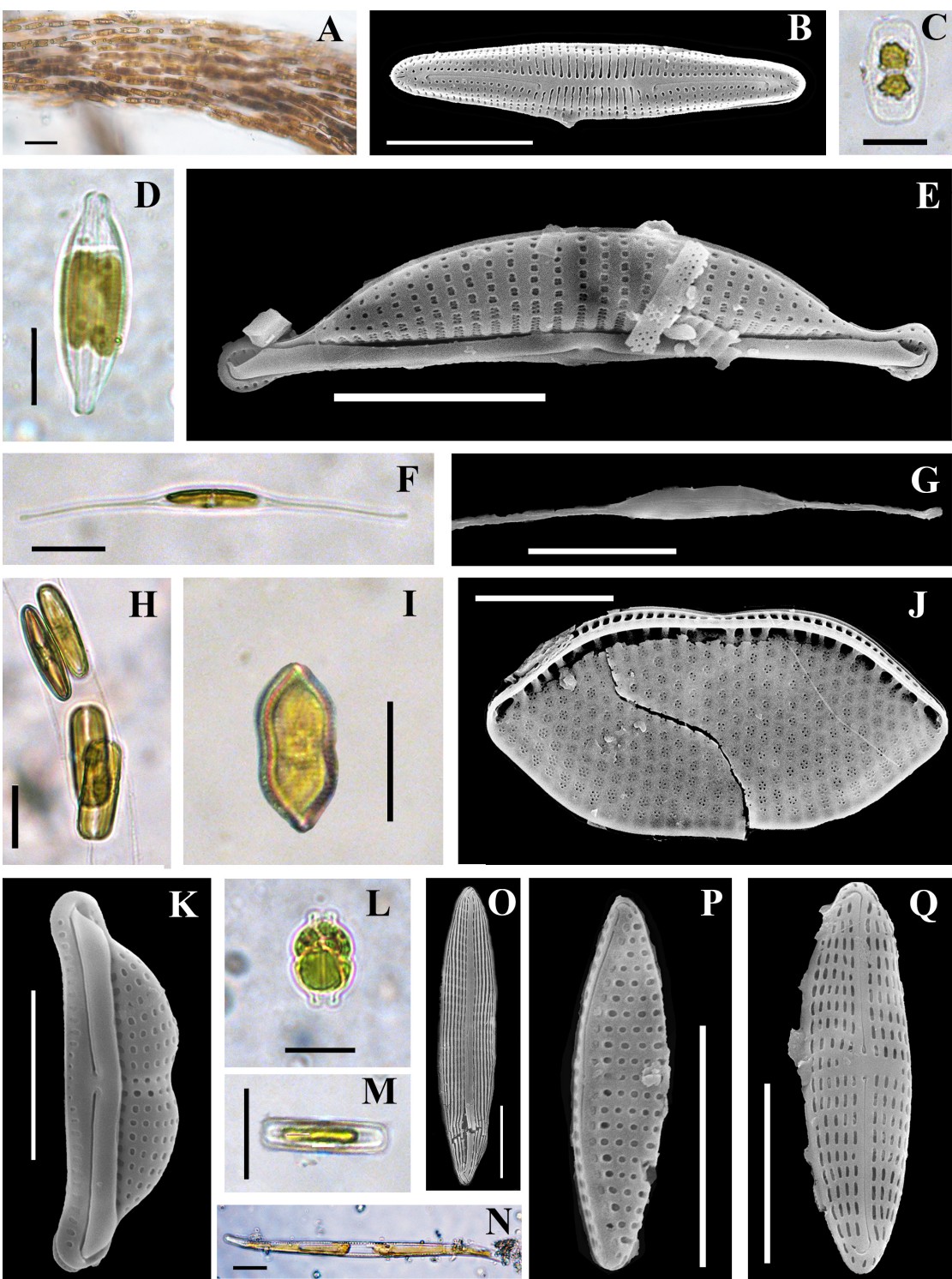

**Figure 6.** Common and frequently occurring diatoms in epilithon in three bays in Sevastopol. (**A,B**) *Berkeleya rutilans*; (**C**) *Undatella lineolata*; (**D,E**) *Halamphora coffeiformis*; (**F,G**) *Cylindrotheca closterium*; (**H**) *Parlibellus delognei*; (**I,J**) *Psammodictyon panduriforme* var. *minor*; (**K,L**) *Amphora bigibba*; (**M**) *Navicula ramosissima*; (**N**) *Nitzschia sigma*; (**O**) *Haslea subagnita*; (**P**) *Nitzschia inconspicua*; (**Q**) *Navicula perminuta*. (**A,C,D,F,H,I,L–N**) = light microscopy; other photos = SEM. Scale bar: (**A,C,F–I,M,N**) = 20 μm; (**D,L**) = 10 μm; (**B,E,J,O,P**) = 5 μm; (**K,Q**) = 4 μm.

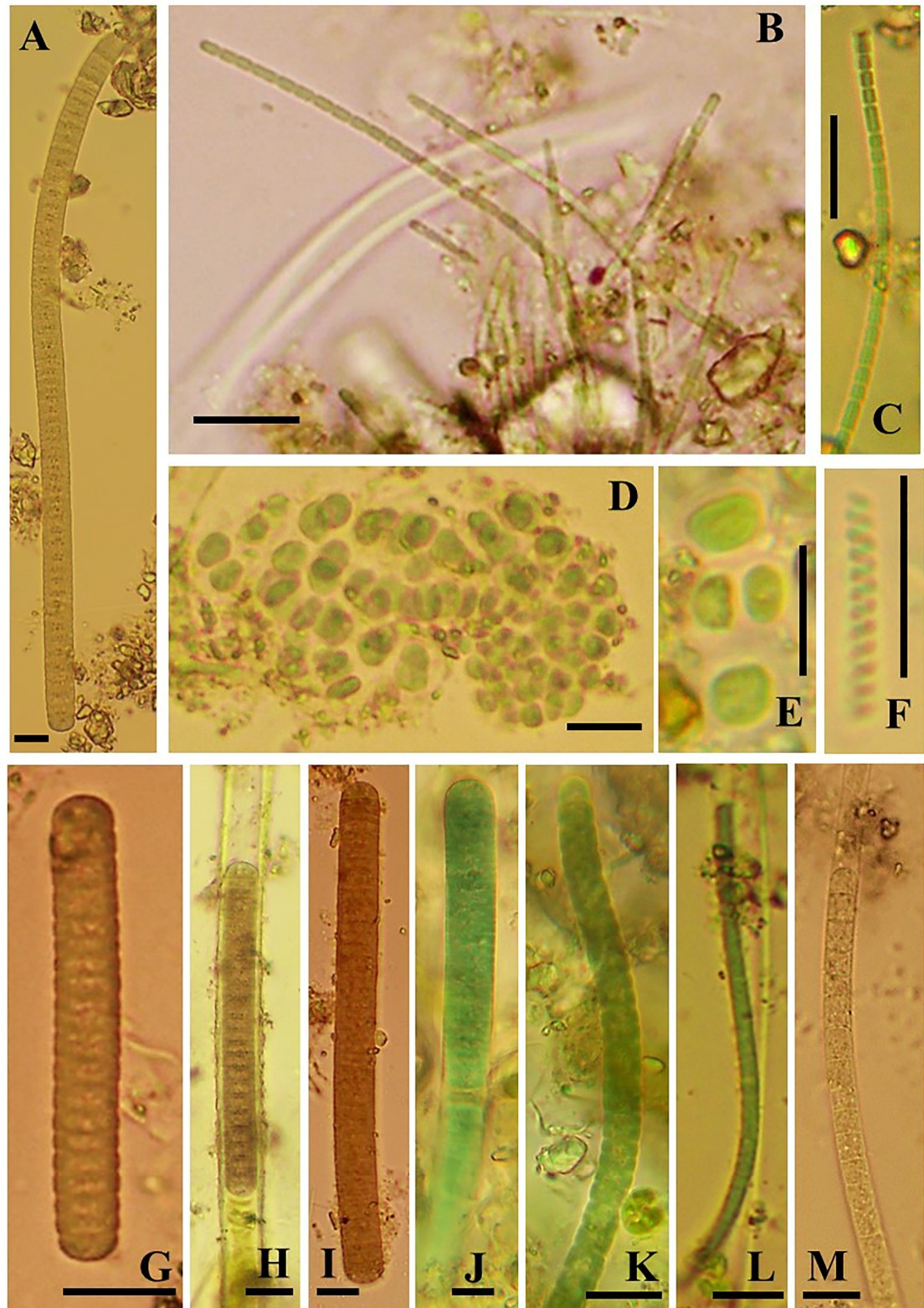

**Figure 7.** Cyanobacteria of epilithon in three bays in Sevastopol. (**A**) *Phormidium nigroviride*; (**B**,**C**) *Leptolyngbya fragilis* (colony and single filament); (**D**) *Gloeocapsopsis crepidinum*; (**E**) Chroococcus minutus; (**F**) *Spirulina subtilissima*; (**G**) *Oscillatoria corallinae*; (**H**) *Lyngbya martensiana*; (**I**,**J**) *Oscillatoria bonnemaisonii*; (**K**) *Phormidium holdenii*; (**L**) *Leptolyngbya subtilis*; (**M**) *Potamolinea aerugineacaeruleum*. Scale bar = 10 μm.

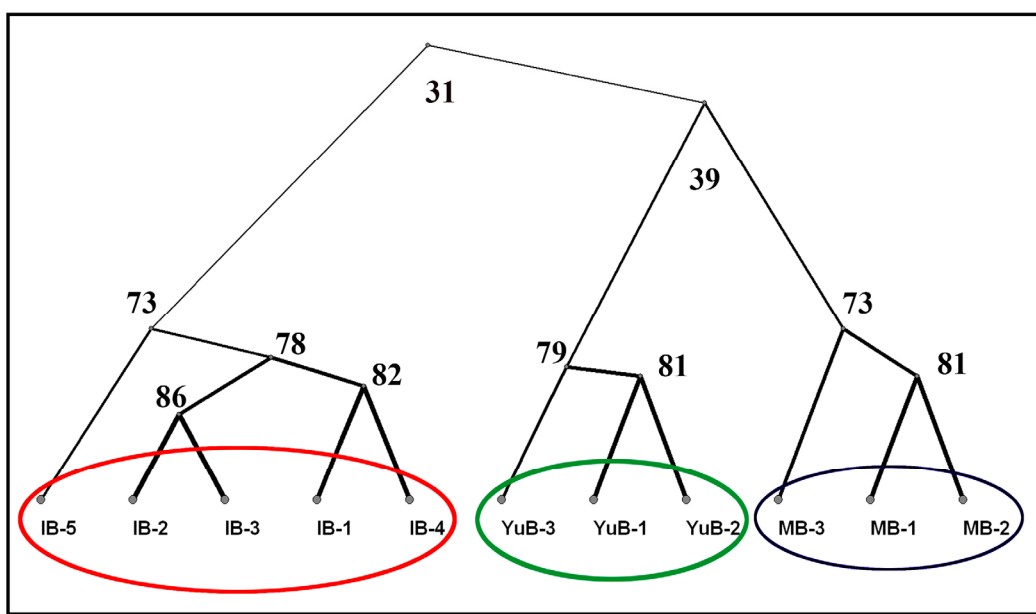

**Figure 8.** Dendrogram of the rock similarity (in %) according to the structure of diatom and cyanobacteria communities; red—Inkerman Bay, green—Martynova Bay, blue—Yuzhnaya Bay.

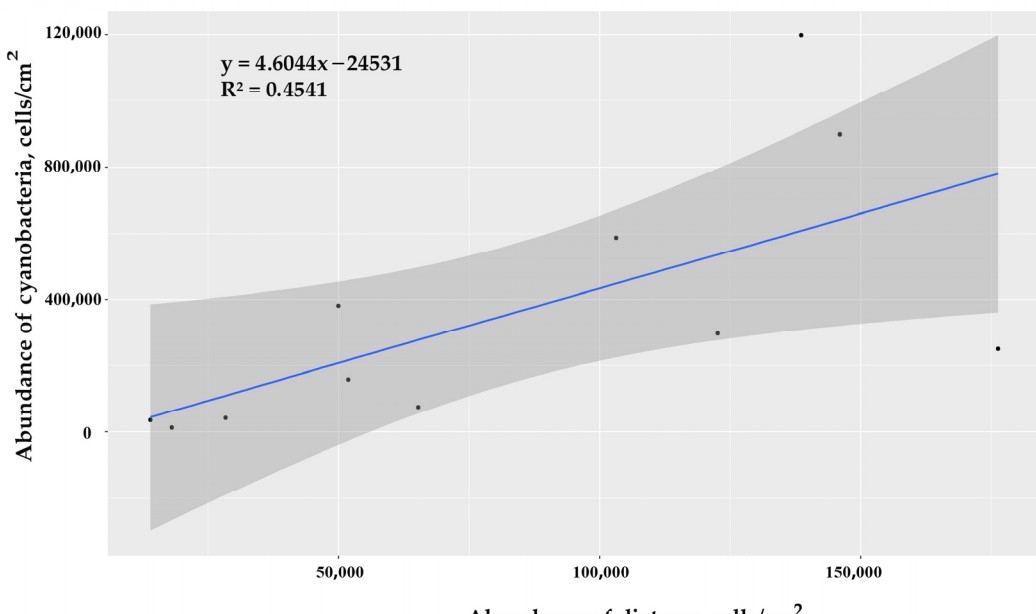

**Figure 9.** Correlation between the total abundance of diatoms and cyanobacteria in the epilithon of three bays in Sevastopol.

Since the cluster analysis of elements in the surface layer of rocks demonstrated the greatest unevenness in the distribution of Si, Ca, and Fe, these three elements were selected to test the effect of their contents on the diatom and cyanobacterial populations' abundances. The results of the correlation analysis for diatoms are shown in Figure 10a. The assessment of correlations showed that the largest number of diatom species (25 out of 39 included in the calculations) had noticeable to very high correlations with the Fe content. The highest association was noted for *Bacillaria paxillifer* and *Navicula directa*. However, despite this, the correlation coefficient between the iron content and the total number of diatoms was only 0.1 (Table 4). Only fourteen species of diatoms showed significant correlations with Si, with nine of them being negative. Almost no significant correlations

were found for Ca. The results of correlation analysis of the element content and the abundance of cyanobacteria are shown in Figure 10b. The cyanobacterial species *S. subsalsa*, *C. minutus*, and *P. minima* from MB had strong positive correlations with Ca ($r_s$ = 0.71–0.73) and negative correlations with *Si* ($r_s$ = 0.71–0.73). Strong negative correlations were found between Fe and the dominant species *L. confervoides* ($r_s$ = −0.94) and *P. aeruginaocaeruleum* ($r_s$ = −0.60) (Figure 7M) from MB, and positive correlations were found with the dominants *P. retzii* ($r_s$ = 0.62), *K. laetevirens* ($r_s$ = 0.71), and *P. holdenii* ($r_s$ = 0.81) (Figure 7K) from IB. Despite the fact that these species were dominant, their dependence on Fe did not affect the results of the correlation analysis of the abundance of the entire cyanobacterial community. As shown in Table 4, the correlation coefficient between iron content and the abundance of the cyanobacterial community was *r* = −0.2.

**Table 4.** Correlation coefficients (r) for the total abundance of diatom and cyanobacteria communities of epilithon and the element contents in the rock surface layers.

| Elements | Diatom Communities | Cyanobacteria Communities |
|:---:|:---:|:---:|
| O | 0.67 | 0.21 |
| C | −0.45 | 0.04 |
| Ca | 0.48 | 0.04 |
| Fe | 0.11 | −0.23 |
| Si | −0.28 | −0.27 |
| Cl | 0.11 | 0.37 |
| Na | −0.01 | 0.32 |
| Mg | 0.19 | −0.23 |
| K | −0.46 | −0.47 |
| Al | −0.42 | −0.55 |
| S | −0.14 | −0.27 |
| Mn | −0.17 | −0.16 |
| Zn | 0.06 | −0.32 |
| Pb | 0.10 | −0.43 |

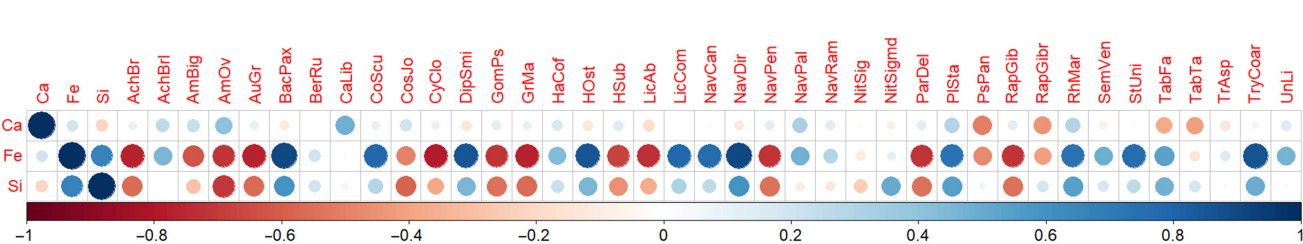

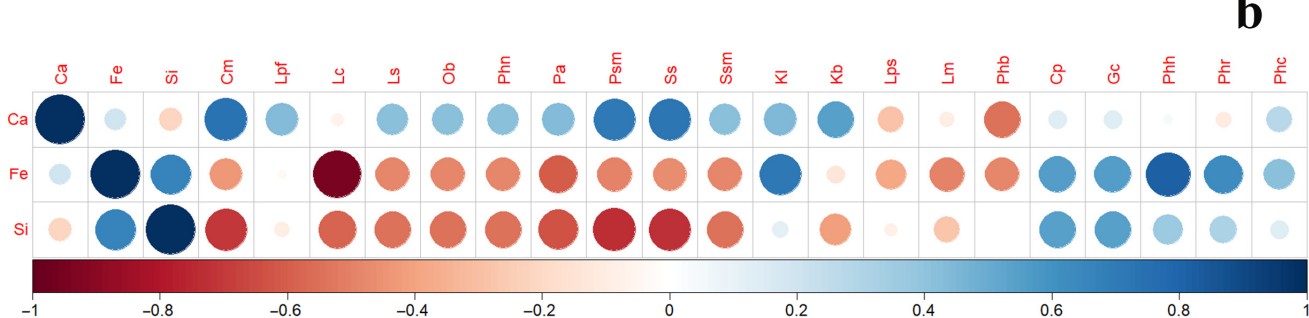

**Figure 10.** Correlations of the abundance of diatoms (**a**) and cyanobacteria (**b**) with the element contents in the rock surface layer.

For 11 diatom species, *Achnanthes brevipes*, *Berkeleya rutilans*, *Caloneis liber*, *Halamphora coffeiformis* (Figure 6D), *Navicula palpebralis*, *Navicula ramosissima* (Figure 6M), *Nitzschia sigma* (Figure 6N), *Seminavis ventricosa*, *Tabularia tabulata*, *Trachyneis aspera*, and *Undatella lineolata*, and one cyanobacterium, *Leptolyngbya fragilis* (Figure 7B,C), no correlations of the abundance with the element contents were noted ($r_s < \pm 0.5$)

## 4. Discussion

Our study is based on the hypothesis of the significant effect of the rock surface elemental composition on the formation of benthic diatoms and cyanobacterial communities and the biocenotic relationships that arise in this interaction. It is known that the rock surface composition reflects the chemical regime of the aquatic environment, and the chemical composition of the seabed, in turn, affects the seawater composition [29], which is consistent with the data obtained. For example, the main mass of carbonates in water is associated with the dissolution of carbonate rocks. In our study, it was shown that the elemental composition of the rock surfaces in three bays differed from each other mainly in the distribution of such elements as Ca, Fe, and Si, which had the highest weight and were dependent on the aquatic environment. This may be due to the physiographic features of Sevastopol Bay, which has a rugged coastline which provides zoning and differences in the thermohaline and hydrochemical characteristics [34].

The registered composition of diatom and cyanobacterial communities in the epilithon of Sevastopol Bay was typical for the Black Sea. Cyanobacteria on rock substrates were not very diverse, and all the species had been found on the Crimean coast earlier [13,52]. The most abundant cyanobacteria in the epilithon samples under consideration were those of the *Lyngbya* and *Phormidium* genera, which are common on rocks [32,53]

For most of the diatom species abundance, noticeable to very high correlations with Fe were found, which are due to the "iron-loving" nature of these microalgae. From data from the literature, the presence of ferritin in diatoms is known, which gives them an advantage over algae that do not contain ferritin [8,30]. It is a very interesting property for biotechnology. In the conditions of Fe deficiency, diatoms have several strategies of reducing the need for this element [54–58], e.g., by forming siderophores—the iron-storing compounds [8]. In addition, adaptations at the genome level may evolve under some circumstances, e.g., in relation to iron bioavailability [59]. In a review paper on the effect of Fe on phytoplankton [60], the authors point out that an increase in the content of iron in water in natural habitats and in cultures positively affects the growth rate, species diversity, and abundance of diatoms. Many diatom species contain high-affinity ferric reductases that dissociate iron III from ligands, multicopper oxidases that oxidize $Fe^{2+}$ to $Fe^{3+}$, and permease systems that receive iron (II) for translocation across membranes [8,61].

The strong negative correlation of Fe with the dominant cyanobacteria from MB that is deficient in iron and the strong positive correlation with the dominants from the iron-rich IB indicate the sensitivity of certain cyanobacterial species to Fe availability. It is known that both the excess and deficiency of iron result in severe abnormal changes in microbial physiology through the generation of reactive oxygen species and free hydroxyl radicals, which ultimately cause cell death [62]. The color of cyanobacteria in the epilithon of MB (Figure 7C,E,F,J,L,M) was more bluish than in species from IB. This may be related to the decrease in cellular pigmentation during iron deprivation causing a blue shift in the main red absorption band of chlorophyll *a* and inducing the iron acquisition process [63]. Iron is also crucial for other essential processes like photosynthesis and respiration [62]. This apparently demonstrates the ability of the species to deposit Fe compounds when iron is deficient in the environment by regulating their metabolism through reducing the need for iron or forming siderophores [64] similar to those in diatoms [8,30]. Apart from siderophores, ferric citrate is another simple compound that has iron-chelating properties and is transported into the cells of cyanobacteria [65]. On the other hand, in the iron homeostasis in *K. laetevirens*, *P. holdenii*, and *P. retzii*, the dominant cyanobacteria in IB, where there is a high concentration of Fe in the water and sediments, the central role is

likely played by bacterioferritin, similar to that in the model cyanobacteria *Synechocystis* sp. [66]. In recent experiments with cultures of cyanobacteria, it was shown that they were able to extract Fe from the solid magnetite phase [67]. Thus, the epilithic cyanobacteria found in IB, *K. laetevirens*, *P. holdenii*, and *P. retzii* [47], may also be able to extract iron from the rock surface.

Diatoms are one of the main consumers of Si in seawater [7,10,26]. However, the correlation coefficients of the diatom abundance and the content of this element in the rock surface layer did not exceed 0.7, and in many species, there was no correlation at all. This can be explained by the fact that diatoms can consume only an aqueous solution of silicic acid $Si(OH)_4$, which is transformed to $SiO_2$ after accumulation in special organelles [7]. It follows from this that diatoms do not consume silicon from the surface of rocks but only from the aquatic environment.

Diatoms are organisms that concentrate Si and Fe [8,27], which are cofactors of many enzymes and affect the formation of nucleic acids [16]. This may be the reason for the largest abundance of diatoms in Inkerman Bay, which is characterized by a high concentration of iron ions in the water [47] and, consequently, the high levels of this element in the seabed [10]. It is known that diatoms are the only microalgae capable of assimilating iron from colloidal forms [10]. It is a very important property for biotechnology.

Ca promotes the secretion of polysaccharides, which ensure the adhesion of attached diatoms to the substrate and the formation of colonies [31]. However, no significant correlation was found between Ca and the diatom abundance. In cyanobacteria, a strong correlation was found between the abundance of *S. subsalsa*, *C. minutus*, and *P. minima* on rocks of continental origin in Martynova Bay and Ca content in the rock surface, which may indicate the ability of these species to biocalcify.

The lack of significant correlations between the element contents and the average abundance of diatoms and cyanobacterial communities can be explained by the stronger influence of other factors on the formation of microphytobenthos communities. For example, there is information about the absence of any significant effect of the chemical composition of rocks on the biomass of benthic cyanobacteria and diatoms in river mouths [68]. As the author points out, the position of the substrates relative to the flow and the flow intensity are much more important factors, and the insignificance of the rock composition effect is accounted for by the low rate of dissolution of the rocks in the experiment. In addition, there are not only individual, but also synergistic effects of elements on the development of diatoms and cyanobacteria. For example, it was found in [69,70] that under Fe limitation, specific growth rates and cell sizes of cultivated diatoms decreased, and after an increase in the Cu content, their growth rates again increased. In the work [71] on four cultivated diatoms, it was shown that by varying the ratios of Na, Ca, and Mg, one can tune the growth rate of a species and achieve its optimum.

In the literature, there are data on the effect of the substrate area and surface roughness on the colonization by benthic organisms. It was shown that diatoms and cyanobacteria prefer substrates with high roughness and an uneven surface [22]. Wahl (1989) described in detail the model of substrate colonization by benthic organisms, in which it was noted that the colonization of surfaces by diatoms may be preceded by the growth of bacteria [72]. Therefore, the elemental composition of rock substrates can have both direct and indirect effects on communities since biofilms always contain organisms capable of leaching metal ions [33].

In our study, the similarity of the elemental composition of rocks in the same area was as high as above 80%, and exactly the same result was obtained when comparing communities in the same area. At the same time, the similarity of rocks from different regions was also quite strong, from 68 to 80%, whereas communities from different areas were similar only by 31–39%. This suggests that the elemental composition was not a decisive factor, and communities are strongly affected by other environmental factors specific to each area. The data obtained show the geochemical inhomogeneity of the habitat of benthic organisms, which is expressed in the mosaic pattern of the benthic communities

but is closely associated with a multitude of other abiotic factors inherent to the study areas. In [73], the authors emphasize that for coastal estuarine ecosystems with high environmental variability, multivariate analysis is required, which would help achieve an integral view on the strength and direction of changes of the environmental variables. At this stage, we have revealed only the effect of the elemental composition of the substrate surface, but in the future, multivariate studies of the influence of environmental conditions on microphytobenthos communities are needed. This is especially relevant to estuarine ecosystems, where one of the main factors affecting microphytobenthos is freshwater runoff, whose effect can be uneven, as demonstrated in the above-mentioned work. Perhaps for this reason, the similarity of communities across the areas was very low: it could be affected by the Chernaya river inflow in IB, sewage discharge in YuB, and seawater recharge in MB.

As for other factors affecting microphytobenthos communities, authors often single out such a factor as seasonality [74–77]. The effect of seasonality was noted also in our previous works on the Black Sea [4,78,79]. This factor encompasses the totality of environmental conditions, such as temperature, salinity, illumination, storms, and many others. Earlier, for the microphytobenthos of the Sevastopol coast, we established a strong correlation between the abundance of microalgae and water temperature [80].

The effect of salinity on microphytobenthos is clear in the example of Sivash Bay in Crimea, in which salinity varied from 30 to 100 psu [81]. It was revealed that the salinity variation is not the only or major factor that determines the species composition of the microphytobenthos. Other factors (total suspended matter concentration, etc.) can play an equally important role. However, quantitative indicators of microphytobenthos may positively correlate with salinity.

Most of the species found on the rocks are cosmopolites, i.e., they are common in the world and live under a wide range of environmental conditions. As for diatoms, for the majority of dominants and species encountered in all regions, a correlation with the elemental composition of the rocks' surfaces was not found. These facts apparently explain why our hypothesis about the effect of the rock surface composition on benthic communities has not been confirmed to as large an extent as expected. It is interesting to note that the number of species found on rocks from YuB in this study was 1.5–2-fold lower than those in IB and MB. The chemogenic origin of the rocks in Yuzhnaya Bay is known, while the limestones in Martynova and Inkerman bays are of biogenic origin [35]. It is also known that more than 80% of carbon-bearing rocks are of precisely biogenic origin. Often, these rocks are formed as a result of the vital activity of cyanobacterial communities and other organisms. Among them are deposits of phytogenic origin—formed by carbonate excretions of microalgae [35]. This may explain the high content of carbon in the elemental composition of rock surfaces and the species-rich communities of microphytobenthos in IB and MB, where the geology of rocks indicates historically more favorable conditions for the development of communities than in YuB, where conditions favored chemogenic processes of rock formation. This study highlights the manifestations of the biogeoecology laws [18] at the biocenotic and population levels. The data obtained show that the mosaic pattern of the rock epilithon is associated with the inhomogeneity of the geochemical environment in the water area. The equal involvement of both cyanobacteria and diatoms was noted in the formation of fouling communities, which may indicate their mutually beneficial interactions and similar response to environmental factors. For example, cyanobacteria produce cellulose, which serves as a defense against stress factors for other organisms [82]. Such connections may have developed historically owing to co-evolution [30]. The work [83] shows the ability of microcommunities to leach metal ions from rock, which are then accumulated in cellulose biofilm, and the effects of rock on the metabolism of microorganisms. These biochemical features emphasize the interrelation between the structure of fouling communities and biomineralization. This means that geomicrobiological studies are important not only in studying ecology and biodiversity but also for solving problems of the impact of microorganisms on the geological environment [84]. We suggest relying on a combination of biology and geoecology methodologies to study different

benthic communities to isolate members of the community for different operational purposes (in biotechnology, biomining, etc.).

## 5. Conclusions

As part of the present study, the geochemical inhomogeneity of the habitat of benthic organisms and the mosaic pattern of epilithic communities on rocky beds have been shown. The data obtained have proven the close connection between the formation of sedimentary rocks and the complex of abiotic factors in the aquatic (marine) environment, which affect the elemental composition of the rock surface. It was shown that the rock surface elemental composition lacked influence on the epilithon communities but was present for populations of some diatom and cyanobacterial species. Perhaps this result was due to the fact that most of the species found on the rocks are cosmopolites. At the same time, we have found several diatom and cyanobacterial species whose population abundance significantly correlated with the iron and calcium contents in the rock surfaces. This expands our knowledge of species that can be used in cultivation and in search of other biotechnologically valuable microbial species. The results have shown the mutually beneficial interactions of cyanobacteria and diatoms in the formation of communities on rocks and proven the relationship between species diversity and the predominance of chemogenic or biogenic processes in the rock formation in different areas.

**Author Contributions:** A.B. and L.R. developed the concept and methodology of this study; D.L. conducted sampling; A.B. and E.M. provided microalgae sample processing; A.B. and D.L. prepared diatom samples and obtained SEM images; L.R. and D.B. identified diatom species; E.M. identified cyanobacterial species; A.B., E.M., D.B. and S.B. conducted data formal analysis; A.B. and S.K. conducted statistical analysis. All authors took part in writing, reviewing, and editing this manuscript. All authors have read and agreed to the published version of the manuscript.

**Funding:** This research was funded by state assignment No. 121030300149-0 from A.O. Kovalevsky Institute of Biology of the Southern Seas of RAS.

**Institutional Review Board Statement:** Not applicable.

**Informed Consent Statement:** Not applicable.

**Data Availability Statement:** All data used in this study are available upon request from the corresponding author.

**Acknowledgments:** We would like to express our gratitude to V.N. Lishaev for the assistance in obtaining SEM images and A.A. Pugach for the assistance in processing the results of this research using modern software. We also thanks to Israeli Ministry of Aliyah and Integration for support.

**Conflicts of Interest:** The authors declare no conflict of interest.

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
