# Peer review of "Does the Elemental Composition of Rock Surfaces Affect Marine Benthic Communities of Diatoms and Cyanobacteria?"

_jmse, doi:10.3390/jmse11081569_

Round 1
Reviewer 1 Report
Dear Authors and Editor,
I went through the manuscript titled ‘Does the elemental composition of rock surface affect marine benthic communities of diatoms and cyanobacteria?’, to be considered for publication on the Journal of Marine Science and Engineering.
Although the idea behind the research is intriguing, the methods applied for investigating the relationship between diatoms, cyanobacteria and rocky substrate are on my opinion very flaw. The authors applied a lot of statistics (on which I am not an expert), but on my opinion the problem relies on the rationale behind the investigation and the analytical approaches, i.e. what precedes the statical analysis.
I suggest to reject the paper, although I strongly encourage the authors to submit again a deeply revised version of the manuscript in the future.
Below I provide some suggestions for ameliorating the work.
Sincerely
Comments
1) I must admit that from the title of your manuscript I expected much more attention to the geology of the Crimean region. Surprisingly, there is a complete lack of geological information in your work. In this regard, are your rock samples pebbles transported from the Chyornaya River into the Sevastopol Bay? If so, you should describe all the possible geological substrates eroded by the river. Are your rock samples in-situ, i.e. do they reflect exclusively the local geological substrate? If so, you should describe in detail the local geology of Sevrastopol Bay. In any case, such introductory considerations are absolutely mandatory on my opinion.
2) How many rocks did you sample? What was your sampling strategy? Did you sample the rocks randomly or on the basis of their different mineralogical composition? On my opinion, this is the only way for better understanding if the elemental composition of a rock surface (which ultimately derives from the interaction between rock minerals, water and organic matter) plays a role in shaping the benthic communities.
3) Line 52 – “The chemical composition of seawater affects the chemical composition of sea rocks”. Reading the manuscript (e.g., 297-305; 342-347 375-376) you often emphasize this aspect, in this way suggesting a one-way relationship (i.e. seawater affecting the composition of rock surface), excluding the possible effect of rock mineralogy on seawater composition… Are you sure that the second relationship can be ignored? How can you test this hypothesis in your specific case study if you sampled the rocks randomly and you don’t know their mineralogical composition?
4) Following the previous comment: a more detailed mineralogical investigation must be performed for each rock sample. This analysis must be based on petrographic observations of rock thin sections and fresh fragments and/or x-ray powder diffraction, and not simply on EDS spectra. I would like to see, in a thin section, how thick is the benthic mat developed on rock surface; I would like to see the different textures of the rock surface that you analyzed, and so on... Of course, this is not a geological paper, but if you want to confirm/exclude the role of rock substrate on the benthic community composition, a precise characterization of your rock record is mandatory on my opinion.
5) How did you prepare the material for SEM-EDS analysis? Did you coat the samples with gold/carbon or you performed the analysis in low-vacuum?
6) The figures 2-3 are not informative: more details at higher magnification are needed.
7) Line 94 – What do you mean with ‘seasonal minerals’?
8) Lines 120-122 – Sorry but this passage sounds very odd to me: what exactly do you mean with rock of marine vs continental “origin”?
9) Lines 372-373 – Maybe you forgot to delete the initial comment… If not, I would say that conclusions are ALWAYS mandatory…
English is ok, but minor revisions are needed
Author Response
Reviewer 1
Dear Editor,
Thank you and the Reviewer 1 for comments. Please find below the point-by-point answers.
With best regards,
Prof Sophia Barinova,
Corresponding author
Dear Authors and Editor,
I went through the manuscript titled ‘Does the elemental composition of rock surface affect marine benthic communities of diatoms and cyanobacteria?’, to be considered for publication on the Journal of Marine Science and Engineering.
Although the idea behind the research is intriguing, the methods applied for investigating the relationship between diatoms, cyanobacteria and rocky substrate are on my opinion very flaw. The authors applied a lot of statistics (on which I am not an expert), but on my opinion the problem relies on the rationale behind the investigation and the analytical approaches, i.e. what precedes the statical analysis.
I suggest to reject the paper, although I strongly encourage the authors to submit again a deeply revised version of the manuscript in the future.
Below I provide some suggestions for ameliorating the work.
Sincerely
Answer. Dear Reviewer, thank you very much for your work aimed at improving the level and quality of our research. We apologize for many incorrect formulations and serious flaws in the methodology presentation.
In the revision, we have expanded the rationale for the study, and this has helped us to apply new analytical approaches. Your comments have allowed us to find new patterns in the data obtained. As a result, it has become clear that it is needed to take into account the whole complex of abiotic factors in each study area, but at the current stage, only multiannual data on the contents of elements in water and detailed characteristics of the geography and geology of the study areas are available to us.
Comments
1) I must admit that from the title of your manuscript I expected much more attention to the geology of the Crimean region. Surprisingly, there is a complete lack of geological information in your work. In this regard, are your rock samples pebbles transported from the Chyornaya River into the Sevastopol Bay? If so, you should describe all the possible geological substrates eroded by the river. Are your rock samples in-situ, i.e. do they reflect exclusively the local geological substrate? If so, you should describe in detail the local geology of Sevrastopol Bay. In any case, such introductory considerations are absolutely mandatory on my opinion.
Answer 1. We possess the information about the geology of the Crimean Peninsula and took it into account when setting up the experiment. This information has been inadvertently overlooked in the previous version of the manuscript, and we thank you for pointing out this deficiency. We added this data to the study rationale and methods, which helped us draw the correct conclusions.
2) How many rocks did you sample? What was your sampling strategy? Did you sample the rocks randomly or on the basis of their different mineralogical composition? On my opinion, this is the only way for better understanding if the elemental composition of a rock surface (which ultimately derives from the interaction between rock minerals, water and organic matter) plays a role in shaping the benthic communities.
Answer 2. We understand that the requirements for the "field" experiments oblige us to make the number of samples much larger than that accepted in this work. However, the complexity of microphytobenthos samples processing allowed us to analyze only 11 rock samples. Sampling was guided by the random sampling method, which is widely used in hydrobiology. At this stage, it was important for us to identify whether there is a general effect of the surface of the stony substrate on the diatoms and cyanobacteria communities and how great it is. We intended to get an answer to a specific question on the links between the element composition in rocks and the diatom and cyanobacterial benthic communities, and we believe that we have achieved the desired goal. However, in the future we will try to apply controlled conditions to find out nuances of this issue.
3) Line 52 – “The chemical composition of seawater affects the chemical composition of sea rocks”. Reading the manuscript (e.g., 297-305; 342-347 375-376) you often emphasize this aspect, in this way suggesting a one-way relationship (i.e. seawater affecting the composition of rock surface), excluding the possible effect of rock mineralogy on seawater composition… Are you sure that the second relationship can be ignored? How can you test this hypothesis in your specific case study if you sampled the rocks randomly and you don’t know their mineralogical composition?
Answer 3. We apologize for the incorrect statement that the chemical composition of seawater affects the chemical composition of marine rocks. You are absolutely right: the cited publications talk about the two-way connections and even specifically about the washing out of carbonates from limestone rocks.
4) Following the previous comment: a more detailed mineralogical investigation must be performed for each rock sample. This analysis must be based on petrographic observations of rock thin sections and fresh fragments and/or x-ray powder diffraction, and not simply on EDS spectra. I would like to see, in a thin section, how thick is the benthic mat developed on rock surface; I would like to see the different textures of the rock surface that you analyzed, and so on... Of course, this is not a geological paper, but if you want to confirm/exclude the role of rock substrate on the benthic community composition, a precise characterization of your rock record is mandatory on my opinion.
Answer 4. Such work is constantly carried out by geologists, and we have tried to apply the known data on the region geology for our tasks. In the methodology, we have refined the characteristics of our rock record. We apologize for having omitted this principal section .
5) How did you prepare the material for SEM-EDS analysis? Did you coat the samples with gold/carbon or you performed the analysis in low-vacuum?
Answer 5. We have expanded description of the methodology of sample preparation for SEM. Samples for the SEM-EDX analysis were sputtered with gold-palladium.
6) The figures 2-3 are not informative: more details at higher magnification are needed.
Answer 6. We apologize for the inconveniences caused in the review. The original artwork has been provided by us to the journal in high resolution. It was expected that element maps would serve for visual perception, and for better understanding and analysis, the element maps are accompanied with the backscattered electron spectrum and the percentage of elements in numbers (Figure 4.).
7) Line 94 – What do you mean with ‘seasonal minerals’?
Answer 7. In Materials and Methods, we have explained the concept of "seasonal minerals", which is used by the authors of the patent for determining the rock surface genesis.8) Lines 120-122 – Sorry but this passage sounds very odd to me: what exactly do you mean with rock of marine vs continental “origin”?
Answer 8. Along with the full elemental composition of the surface by SEM-EDS, we also performed a surface genesis analysis using the method presented in the patent. Perhaps the use of this method was superfluous, since we did not get significant results, and it was admissible not to mention it in the manuscript.9) Lines 372-373 – Maybe you forgot to delete the initial comment… If not, I would say that conclusions are ALWAYS mandatory…
Answer 7. Sorry, this is a mistype. This passage from the journal template has been deleted.
Comments on the Quality of English Language
English is ok, but minor revisions are needed

Reviewer 2 Report
PLEASE SEE ATTACHED FILE

ENGLISH FINE. MAYBE AN AMBIGUOUS WORD OR TWO
Author Response
Reviewer 2
Dear Editor,
Thank you and the Reviewer 2 for comments. Please find below the point-by-point answers.
With best regards,
Prof Sophia Barinova,
Corresponding author
The authors thank the Referee for the great work done to improve our manuscript. We appologize for the possible errors and awkward wordings, which may have been difficult to understand.
Comment: “THE TITLE promises an interesting contribution that should be published. However, it
provides mainly data on correlative values that do not support a hypothesis. I suggest that
the authors search for such a hypothesis, which I’m sure they had in mind, and state it. It
will upgrade their scientific approach.”
Moreover, the answer to the question in the title is not presented anywhere; not in the
discussion nor in conclusion sections. The answer to the posed question (post-facto
hypothesis) and conclusion should be included at the end of the ABSTRACT.
Answer: We stated the hypothesis of diatom and cyanobacterial benthic сomunities being primarily affected by the difference in the elemental composition of the substrate. As a result of our work, the underlying hypothesis has not been confirmed. The elemental composition of the rock surface turned out to be not a decisive factor affecting the abundance of benthic diatom and cyanobacterial communities. This statement has been added to the Abstract as proposed. We thank the Reviewer for this remark, and we have included the hypothesis in the manuscript. However, some correlations have been found between the elemental composition of rock surfaces and abundance of certain species of diatoms and cyanobacteria, which result has been also reflected in the manuscript.
Comment: “LINES 29-32: To my appreciation, these lines should be eliminated. The introduction will start fine in the next line.”
Answer: With these lines, we wanted to introduce the reader to the fundamental area of research. This is because the study of rocky beds of marine ecosystems and the communities inhabiting them was previously studied unilaterally, either geologically or biologically. We emphasize that the environment and the organism in the sea are interconnected by biogeochemical cycles, and therefore it is important to consider them as a whole.
Comment: “LINES 58-60: “The aim of the woaark is to study the formation of communities of diatoms and cyanobacteria on rocky seabeds in relation to the elemental composition of the rock surface in three bays in the Black Sea.”
According to the title this would not be the pursued aim or objective, but to determine the
structure of the diatom and cyanophyte taxocoenoses (or assemblages/associations) in
relation to the elemental composition of the rock surface...on the basis of species
composition and cell densities…
Although the lacking hypothesis (pronosticated answer) should state “the elemental
composition of rock surface affects the species composition of benthic diatom and
cyanophyte taxocoenoses” This hypothesis would have to be preceded by a premise, e.g.,
Because availability of certain elements or minerals have been observed to influence diatom
and cyanophyte taxocoenoses in a way that….I RECOMEN TO INCLUDE THIS
RIGHT AFTER PROPOSING THE AIM OR OBJECTIVE IN THE INTRODUCTION OTHER OBSERVATIONS
Answer: We set the goal of getting an answer to the specific question on the microphytobethos community structure and we believe we have achieved this goal. We succeeded in proving the close relationship of the elemental composition of the surface of sedimentary rocks and the aquatic environment conditions of the area. In addition, this study allowed us to identify individual species that have specific features, and in the future we plan to study this issue in controlled conditions.
Comment: “LINES 169-170. It is not formal statistics to present mean values for a nominal scale; try median or modal values, or just the intervals. Mean values of abundances do not have real
meaning.”
Answer: They were replaced with intervals in which the number of diatoms varied in each of the bays.
Comment: “Table 2 shows an amazingly familiar cast of diatom taxa that are widely distributed
worlwide. This observation may be used to widen the reach of the inferences
Answer: Thanks for the valuable remark, we used this fact in the manuscript to justify the fact that our hypothesis was rejected as a result of the work.
Comment: “PLATE Figure 6o Nitzschia sigma is most likely identified. It is to small to be N. sigma, and the chloroplasts seem more like the ones in Gyrosigma (cf. parvulum).”
Answer: In the manuscript, we provided perhaps a not very good photo of this species, and its chloroplasts could have changed after fixation. Figure 6o has been replaced with a higher quality image of Nitzschia sigma. This diatom is 38-1000 µm long, 4-26 µm broad, fibulae (3)7-12 in 10 µm, striae 15-38 in 10 µm (Diatom analysis, 1950; Witkowski et al., 2000). The species found in our study is fully consistent with this description. For example, the specimen presented in the initial version of the manuscript had the sizes 62 x 6 µm, 22 striae in 10 µm. We are very grateful to the Reviewer for this valuable comment and will further pay closer attention to species similar to Gyrosigma parvulum for more accurate identification.
Comment: “DISCUSSION.
Authors should consider that several researchers have observed that benthic diatoms show
a “patchy distribution” (McIntire , C. D. & W. S. Overton. 1971. Distributional patterns in
assemblages of attached diatoms from Yaquina Estuary, Oregon. Ecology, 52(5):758-777.
DOI:10.2307/1936024), and that their findings may be on to providing an explanation for
such aggregation on the basis of elemental composition of the substrate.”
Answer: The authors are very grateful for the useful source in their work. It has been included in the current revision.
Comment: “LINES 296-297: The statement: “It was shown that the chemical composition of the rock surface in the bays differed from each other and was dependent on the aquatic environment. The rock surface composition reflects the chemical regime of the aquatic environment [26], which is consistent with the data obtained” is actually a colateral conclusion or corollary, which
should also be transferred to the conclusions section.
Answer: мы полностью согласны с рецензентом и внесли это утверждение в выводы.
Comment: “LINES 306-309. Why does the presence of ferritin in diatoms would explain the high
correlation found? The explanation is not evident!”
Answer: We have largely supplemented the Discussion section and edited this sentence.
Comment: “LINES 311-317: Also, when they mention that “Many diatom species contain high-affinity ferric reductases that dissociate iron III from ligands, multicopper oxidases that oxidize Fe 2+ to Fe 3+ , and permease (systems that receive iron (II) for translocation across membranes [8, 57]. This information does not explain either why the correlation was high! SOMETHING HAS
TO BE CONCLUDED, I.E., AN OBSERVATION BASED ON THIS FACT.LINES 327-328: LIKEWISE, when they say “This can be explained by the fact that diatoms can consume only aqueous solution of silicic acid Si(OH) 4 , which is transformed to SiO2 after accumulation in special organelles [7]. LIKEWISE, SOMETHING HAS TO BE CONCLUDED
Answer: We have edited these lines and added a little more information. We hope that we have clarified these issues in the revision. Regarding the correlation and discussion on the dependence on iron, we just wanted to present the possible mechanisms that are currently of interest to biotechnologists when looking for model species with potentially useful properties.
Comment: “LINES 354-370: Although this paragraph is well written I suggest that it should be located much earlier in the discussion section (for starters) in substitution of the first four lines.”
Answer: We have significantly redesigned the Discussion section; we hope now this section has become more informative and understandable.
Comment: “CONCLUSION
Although the conclusion section is not mandatory, the authors have exercised their option.
Unfortunately what they present are not conclusions. To my appreciation, what is
presented as conclusion as it is should be eliminated. Conclusions are of deductive or
abductive (hypothesis) logical structure, not a summary of observations (inductive).
If a pre-facto hypothesis had been posed, the conclusion should address the outcome of
contrasting said hypothesis, i.e., refuted or backed, explaining why...for example, because
cyanophyte growth is favored by carbonate availability (they prefer alkaline conditions);
or... iron seems to deter cyanophyte growth while favoring diatoms. Said conclusions are of
course hypothesis (post-facto), which are actually include in discussion and should be
remarked by the authors in the conclusions section.
The main conclusion should deal with the answer to the posed question in the title, and the
explanation or consequences of said answer.”
Answer: We agree with the Reviewer and have completely reworked the Conclusions section.

Reviewer 3 Report
The authors of this study explore an important research question that address the relationship between the biotic community of marine biofilms and the substrate geochemistry. This has not been well-investigated, from my experience, and there is likely a general assumption that the substrate might not matter much. I think the concept that the authors explore is interesting, and of merit. The sampling and sample processing methods, though, are not well-described, and lack detail. The authors refer readings to a publication that is likely not available widely. For instance, it is unclear how many samples were collected from each site, how the rocks were scraped for the phytobenthos, how the samples were preserved, how the diatom samples were processed, how permanent slides were prepared, how many diatom valves/cells were counted (diatoms don't have shells, they have cells, consisting of two valves). These are standard methodological considerations that need to be described more completely and more clearly. A reader can't easily assess whether the results are significant unless they know what part of the study was qualitatively and quantitatively done. The authors show some excellent SEM and light micrographs of living diatom cells. But, diatoms can not be identified without removing the organic matter inside the cells first, because the striae count and pattern are needed for identification purposes. I would suggest that the methodology for the biotic component of this project be rewritten to identify the above details. These details will help in understanding if the differences in abundances, concentrations, etc. are meaningful or important. It was also unclear why sea water was taken for sample processing. That isn't typically needed, and would likely contain phytoplankton species which could influence the analysis of the benthos.
I think the interpretation of the results appears reasonable but perhaps modest, and but that the authors should discuss a bit more the other factors that can influence algal communities - salinity, temperature, nutrients, and the presence of grazers. Those should be mentioned in the discussion, and if any data are available for these during sampling, they should be provided in the study site description. The authors recommend that this type of study could provide information for different operational purposes, and I think that part of the study needs to be explained more clearly.
The paper is generally well-written.
Author Response
Reviewer 3
Dear Editor,
Thank you and the Reviewer 3 for comments. Please find below the point-by-point answers.
With best regards,
Prof Sophia Barinova,
Corresponding author
Comments and Suggestions for Authors
The authors of this study explore an important research question that address the relationship between the biotic community of marine biofilms and the substrate geochemistry. This has not been well-investigated, from my experience, and there is likely a general assumption that the substrate might not matter much. I think the concept that the authors explore is interesting, and of merit. The sampling and sample processing methods, though, are not well-described, and lack detail. The authors refer readings to a publication that is likely not available widely. For instance, it is unclear how many samples were collected from each site, how the rocks were scraped for the phytobenthos, how the samples were preserved, how the diatom samples were processed, how permanent slides were prepared, how many diatom valves/cells were counted (diatoms don't have shells, they have cells, consisting of two valves). These are standard methodological considerations that need to be described more completely and more clearly. A reader can't easily assess whether the results are significant unless they know what part of the study was qualitatively and quantitatively done. The authors show some excellent SEM and light micrographs of living diatom cells. But, diatoms can not be identified without removing the organic matter inside the cells first, because the striae count and pattern are needed for identification purposes. I would suggest that the methodology for the biotic component of this project be rewritten to identify the above details. These details will help in understanding if the differences in abundances, concentrations, etc. are meaningful or important. It was also unclear why sea water was taken for sample processing. That isn't typically needed, and would likely contain phytoplankton species which could influence the analysis of the benthos.
I think the interpretation of the results appears reasonable but perhaps modest, and but that the authors should discuss a bit more the other factors that can influence algal communities - salinity, temperature, nutrients, and the presence of grazers. Those should be mentioned in the discussion, and if any data are available for these during sampling, they should be provided in the study site description. The authors recommend that this type of study could provide information for different operational purposes, and I think that part of the study needs to be explained more clearly.
Comments on the Quality of English Language
The paper is generally well-written.
Answer:
The authors are very grateful to the Referee for reviewing the manuscript and for very useful suggestions for improving its quality. We have significantly expanded Section 2 (Materials and Methods), especially as regards the biotic component. We have added a description of sampling and sample processing, sample preparation for SEM and formula for counting the number of cells. Seawater samples were taken from the same depth as the rock samples and delivered to the laboratory for washing off microphytobenthos from the substrate. The water was preliminarily filtered through a 0.45-μm filter.
Section 4 (Discussion) has been also significantly expanded.
The communities of microphytobenthos in marine areas, especially if it is estuarine and under anthropogenic influence, can be simultaneously influenced by a very large number of factors, both abiotic and biotic. However, in this work, we primarily focused on the effects of the elemental composition of substrate surface on the epilithic diatoms and cyanobacteria. Data on the influence of season, temperature and salinity on microphytobenthos are given. Some factors are currently very poorly studied for myсrophytobenthos, such as grazing. In future works of this kind, we will try to carry out a multivariate analysis.

Round 2
Reviewer 1 Report
Dear Prof. Barinova,
I acknoweldge your efforts in providing a revised version of the manuscript. Nevertheless, I must ask you to be much more precise and explicit. Let's say that I am a curious reader and I find your work on the web... The title is intriguing: "Does the elemental composition of rock surface affect marine benthic communities of diatoms and cyanobacteria?". In the abstract I read:
1) "For the majority of the diatom species, correlation with Fe were noticeable to very high (...) For the cyanobacteria (...) strong positive correlation with Ca and negative correlation with Si were observed"
2) "In general, it was found that the elemental composition of the rock surface is not a decisive factor affecting the abundance of benthic diatoms and cyanobacterial communities"
To be honest, as a generic reader, I would struggle a bit in understanding... Firstly you said that there is a correlation between diatoms and Fe and an anticorrelation between cyanobacteria and Si, then you said that rock surface composition does not play a role in shaping the benthic assemblages... Don't you think that this sounds a bit contradictory?
Reading the paper, this passage is rarely clarified.
Moreover, at lines 479-481 you state that "(...) the chemical composition of the rock surface in the bays differed from each other", while at lines 613-617 you state that "the similarity of the chemical composition of rocks in the same area was as high as above 80% (...) At the same time, the similarity of rocks from different regions was also quite strong, from 68% to 80%".
You are probably on the right track to improving your work, but very major revisions are still necessary for publishing it. At the present state the manuscript is quite confusing (see for example lines 648-657: a very puzzling passage!).
I suggest to resubmit the paper after a deep revision, carefully evaluating all those statements that are critical for your work. Don't be in too much of a hurry... Take your time and try to provide a clearer version of this manuscript.
Sincerely
English must be improved.
Author Response
Responses to Reviewer 2-2
Dear Reviewer 2,
Thank you for your comments. We have modified ms with respect of your comments. Please find below the responses to each of your comments.
With best regards,
Prof Sophia Barinova,
Corresponding author
Comments and Suggestions for Authors
Dear Prof. Barinova,
I acknoweldge your efforts in providing a revised version of the manuscript. Nevertheless, I must ask you to be much more precise and explicit. Let's say that I am a curious reader and I find your work on the web... The title is intriguing: "Does the elemental composition of rock surface affect marine benthic communities of diatoms and cyanobacteria?". In the abstract I read:
1) "For the majority of the diatom species, correlation with Fe were noticeable to very high (...) For the cyanobacteria (...) strong positive correlation with Ca and negative correlation with Si were observed"
2) "In general, it was found that the elemental composition of the rock surface is not a decisive factor affecting the abundance of benthic diatoms and cyanobacterial communities"
To be honest, as a generic reader, I would struggle a bit in understanding... Firstly you said that there is a correlation between diatoms and Fe and an anticorrelation between cyanobacteria and Si, then you said that rock surface composition does not play a role in shaping the benthic assemblages... Don't you think that this sounds a bit contradictory?
Reading the paper, this passage is rarely clarified.
Moreover, at lines 479-481 you state that "(...) the chemical composition of the rock surface in the bays differed from each other", while at lines 613-617 you state that "the similarity of the chemical composition of rocks in the same area was as high as above 80% (...) At the same time, the similarity of rocks from different regions was also quite strong, from 68% to 80%".
You are probably on the right track to improving your work, but very major revisions are still necessary for publishing it. At the present state the manuscript is quite confusing (see for example lines 648-657: a very puzzling passage!).
I suggest to resubmit the paper after a deep revision, carefully evaluating all those statements that are critical for your work. Don't be in too much of a hurry... Take your time and try to provide a clearer version of this manuscript.
Sincerely
Comments on the Quality of English Language
English must be improved.
Answer
Dear Reviewer,
The authors are very grateful to you for your attention to our manuscript and for your efforts aimed to show us how to improve it. We really appreciate it. We have revised the manuscript in accordance with your comments and tried to reword the confusing passages in a more understandable manner. In addition, we would like to provide a little bit more explanation of our results.
The first step in the analysis of the rock surface element composition effect on the communities of diatoms and cyanobacteria was a correlation analysis of the total average abundance of communities for each rock.
At the second step, we analyzed the correlation of the abundance of each individual species separately instead of the communities as a whole, e.g. the abundance of populations. At this stage, significant correlations were detected for some species.
In response to your first comment:
1) "For the majority of the diatom species, correlation with Fe were noticeable to very high (...) For the cyanobacteria (...) strong positive correlation with Ca and negative correlation with Si were observed" - This statement refers to the results of the correlation analysis of individual diatom and cyanobacterial population abundance.
2) "In general, it was found that the elemental composition of the rock is not a decisive factor affecting the abundance of benthic diatoms and cyanobacterial communities"
- This statement refers to the results of a correlation analysis of the total abundance of the diatom and cyanobacterial communities.
Thus, it was shown that there is an absence of the rock surface element composition effect on the epilithon communities, but its presence for some diatom and cyanobacterial populations.
We are very grateful for this comment. We have made clarifications in Abstract, Results and Discussion sections.
The similarity of the rock surface element composition was 68-80%. According to the dendrogram in Fig. 5a, the rocks were divided into clusters by regions with a similarity of 80-85%. The differentiation was based on the three elements Fe, Cu, Ca, which allowed us to identify some species of diatoms and cyanobacteria that correlate with these elements. We agree with the Reviewer that this issue was confusingly described in the discussion. Therefore, we have added a clarification in Results and in the Discussion section in.
Lines 648-657 were rewritten. These sentences were aimed to demonstrate the interesting fact that the species diversity of the communities was higher in the bays, where, according to the literature data, the rocks are of biogenic origin.
We tried to improve the English by rephrasing the difficult-to-understand sentences throughout the manuscript.

Round 3
Reviewer 1 Report
Dear Authors,
I noticed your efforts in providing an improved version of the manuscript.
I am very looking forward to see further developments in this intriguing field of research.
At this point, I only suggest very minor revisions.
Sincerely
Last comments:
1) scale bars in Fig. 2 must be enlarged
2) lines 234-236: are you sure that rock surface are useful in this work?
3) line 240: your rock samples have been crushed/splitted (I suppose) before SEM analysis; so specify that you have analyzed only a small portion of your rock samples
4) lines 300, 399 and Tab. 4: oxygen content cannot be reliably calculated by means of EDS analysis. Consider to delete...
5) Fig. 3: EDS spectra are not clear, enlarge them!
Minor revisions of English are needed